# Experimental evidence that group size generates divergent benefits of cooperative breeding for male and female ostriches

**Julian Melgar[1]\*, Mads F Schou[1], Maud Bonato[2], Zanell Brand[3], Anel Engelbrecht[3], Schalk WP Cloete[2,3], Charlie K Cornwallis[1]\***

[1]Department of Biology, Lund University, Lund, Sweden; [2]Department of Animal Sciences, University of Stellenbosch, Stellenbosch, South Africa; [3]Directorate Animal Sciences, Western Cape Department of Agriculture, Elsenburg, South Africa

**Abstract** Cooperative breeding allows the costs of parental care to be shared, but as groups become larger, such benefits often decline as competition increases and group cohesion breaks down. The counteracting forces of cooperation and competition are predicted to select for an optimal group size, but variation in groups is ubiquitous across cooperative breeding animals. Here, we experimentally test if group sizes vary because of sex differences in the costs and benefits of cooperative breeding in captive ostriches, *Struthio camelus,* and compare this to the distribution of group sizes in the wild. We established 96 groups with different numbers of males (1 or 3) and females (1, 3, 4, or 6) and manipulated opportunities for cooperation over incubation. There was a clear optimal group size for males (one male with four or more females) that was explained by high costs of competition and negligible benefits of cooperation. Conversely, female reproductive success was maximised across a range of group sizes due to the benefits of cooperation with male and female group members. Reproductive success in intermediate sized groups was low for both males and females due to sexual conflict over the timing of mating and incubation. Our experiments show that sex differences in cooperation and competition can explain group size variation in cooperative breeders.

**\*For correspondence:**
julian.melgar@biol.lu.se (JM);
charlie.cornwallis@biol.lu.se
(CKC)

**Competing interest:** The authors declare that no competing interests exist.

## Editor's evaluation

This article should be of interest to researchers working on animal behaviour and the evolution of cooperation. It experimentally investigates the effect of differences in group size and group composition on reproductive behavior, using an impressive sample of semi-wild populations of ostriches. Overall, the article offers a valuable analysis of the breeding ecology of ostriches and may inspire similar empirical work on other systems, examining cooperation, group living and breeding ecology.

## Introduction

A key feature influencing the social organisation of cooperative breeding animals is group size (*Alexander, 1974*; *Bourke, 1999*; *Clutton-Brock, 2021*; *Kappeler, 2019*; *Koenig and Dickinson, 2016*; *Rubenstein and Abbot, 2017*; *Taborsky et al., 2021*). In large groups, greater opportunities for cooperation can increase individual reproductive success, for example, by spreading the burden of offspring care among group members (*Alexander, 1974*; *Koenig and Dickinson, 2016*; *Rubenstein and Abbot, 2017*; *Taborsky et al., 2021*). However, as group size increases, more intense competition

**eLife digest** Being a parent is hard work. The unrelenting demand for food and protection is exhausting. Now imagine being a parent on the hot African savannah. Food and water are scarce, and whenever you leave your offspring, they overheat, or something eats them. This is the reality for ostriches. They, like humans, cope with the challenges of parenthood by sharing childcare responsibilities.

Ostriches live in groups, breed in a communal nest, and take it in turns to incubate their eggs. This helps to maximize the survival of their offspring, but it has its downsides. The bigger a group gets, the more its members have to compete over mates and space for their eggs in the nest. The balance between cooperation and competition should, in theory, result in one 'optimal' group size. But this pattern does not seem to hold true: in the wild, ostrich families vary wildly in size and composition.

To find out why, Melgar et al. set up dozens of groups of breeding ostriches and gave them different opportunities to cooperate. For males, there was one group size that maximized the number of offspring they produced (reproductive success): a single male with four or more females. Males did not benefit much from cooperation, and suffered greatly from competing with other males for mates. For females, however, the story was different. They benefited much more than males from cooperation and did best in bigger groups where they could share egg care with other individuals. Middle-sized groups were not good for either sex because reproduction was hard to coordinate: males continued to pursue copulations after females had initiated incubation, resulting in eggs being exposed and broken. The different priorities of males and females explain why there is no single optimal group size for ostriches.

How groups balance competition and cooperation is a fundamental question in biology. Why do some organisms prefer to live alone, while others thrive in large groups? Understanding more about the balance of priorities within a group could hold the answers. It could also help to inform conservation work and animal breeding by showing how different social pressures influence breeding success.

and lower marginal benefits of cooperation can reduce reproductive success, especially in cooperative breeding systems where all group members attempt to breed (*Figure 1A*. *Clutton-Brock, 2021*; *Krause and Ruxton, 2002*; *Powers and Lehmann, 2017*; *Riehl, 2011*; *Vehrencamp, 1983*). Coordinating collective activities can also be more difficult in larger groups leading to the breakdown of group cohesion (*Krause and Ruxton, 2002*; *Focardi and Pecchioli, 2005*; *Papageorgiou and Farine, 2020*).

The changes in cooperation and competition that occur as groups increase in size are predicted to result in an optimal group size (*Figure 1A–B*; *Giraldeau and Gillis, 1985*; *Krause and Ruxton, 2002*; *Markham et al., 2015*; *Powers and Lehmann, 2017*; *Pulliam and Caraco, 1984*; *Yip et al., 2008*). However, in wild populations cooperative breeding groups are extremely variable in size, differing in their numbers of males and females (*Alexander, 1974*; *Clutton-Brock, 2021*; *Koenig and Dickinson, 2016*; *Lott, 1991*; *Lukas and Clutton-Brock, 2018*; *Rubenstein and Abbot, 2017*; *Rudolph et al., 2019*). A common explanation for why group size varies is that fluctuating ecological conditions shift the optimal size of groups over time and space (*Koenig, 1981*; *Yip et al., 2008*; *Zöttl et al., 2013*). Changes in ecological conditions clearly have important effects on groups, but they do not explain why the composition of social groups is often highly variable under similar ecological conditions (*Koenig and Dickinson, 2016*; *Rubenstein and Abbot, 2017*; *Davies, 1992*; *Davies et al., 1995*; *Lukas and Clutton-Brock, 2018*; *Vehrencamp, 1977*; *Hellmann et al., 2015*; *Markham et al., 2015*; *Papageorgiou and Farine, 2020*).

A possible explanation for why groups vary independently of ecological conditions is that there are advantages and disadvantages to being in large and small groups that broaden the range of group sizes where reproductive success is maximised (*Alexander, 1974*; *Davies and Houston, 1986*; *Santos et al., 2015*; *Ferrari et al., 2019*). For example, in small groups sexual competition is low, but there are fewer opportunities for cooperating over offspring care. Conversely, in larger groups the higher costs of sexual competition may be offset by the benefits of cooperative offspring care, resulting in similar reproductive payoffs from being in small and large groups (*Davies et al., 1995*; *Davies, 1985*; *Riehl, 2011*). The changes in the strength of sexual competition and the benefits of cooperation

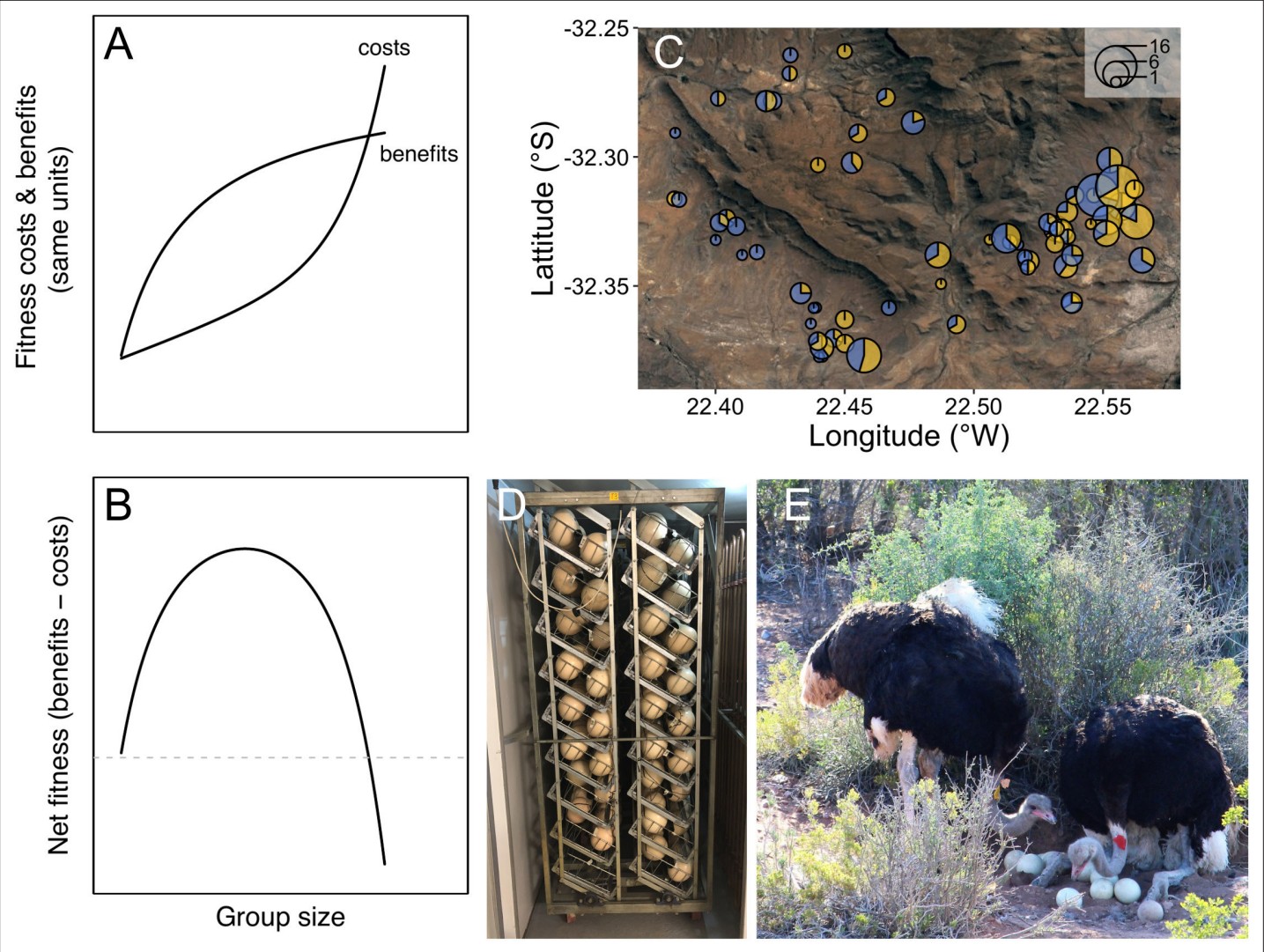

**Figure 1.** Is there an optimal group size for cooperative breeding ostriches? Theoretically, accelerating costs of competition and diminishing benefits of cooperation are expected to result in a single optimal group size (**A–B**, modified from *Krause and Ruxton, 2002*). (**C**) Groups in natural populations are, however, highly variable in size: A satellite image of the Karoo National Park with different groups of ostriches plotted. The size of the circles indicates the number of individuals and the blue and yellow segments indicate the proportion of males and females, respectively. To understand natural variation in group size, experiments that manipulate both the number of males and females in groups, and the benefits of cooperation are required. Opportunities for cooperative incubation were manipulated at the experimental study site by collecting and artificially incubating eggs for part of the breeding season (**D**). Patterns of reproductive success in groups of different size when cooperation was restricted were compared to situations where opportunities for cooperative incubation, such as among these males (**E**), were allowed by leaving eggs in nests (photo by Julian Melgar).

The online version of this article includes the following figure supplement(s) for figure 1:

**Figure supplement 1.** Group size and composition of wild ostrich breeding groups in Karoo National Park in relation to patterns of reproductive success measured under experimental conditions.

that occur when group size varies may also influence males and females differently (*Davies, 1989*; *Lessells, 2012*; *Trivers, 1972*; *Wong et al., 2012*; *Snijders et al., 2021*). The divergent reproductive interests of the sexes can lead to different optimal group sizes and variation in group sizes can occur because there are different outcomes of ongoing sexual conflict (*Davies, 1989*). Understanding group size variation therefore requires accurately measuring the costs and benefits to males and females of breeding in groups of different size (*Sibly, 1983*; *Giraldeau and Gillis, 1985*; *Krause and Ruxton, 2002*; *Rubenstein and Abbot, 2017*).

Empirical estimates of the costs and benefits of group size are typically inferred by measuring the outcome of breeding events, for example, the number of offspring raised to independence

by groups under natural conditions (*Alexander, 1974*; *Koenig and Dickinson, 2016*; *Rubenstein and Abbot, 2017*; *Taborsky et al., 2021*). With such observational data it is not possible to separate the effects of competition and cooperation on reproductive success. Estimates of the advantages of being in certain groups can also be biased by individual differences (*Cockburn et al., 2008*; *Dickinson and Hatchwell, 2004*; *Downing et al., 2020*; *Schoepf and Schradin, 2012*). For example, the benefits of being in large groups can be overestimated if individuals with high reproductive success are more likely to be in larger groups. It is therefore difficult to quantify the benefits and costs to individuals of being in different sized groups from observational data on naturally formed groups.

Experiments are needed that control for ecological conditions, that separate the effect of individual differences from group attributes on reproductive success, and disentangle how competition and cooperation change with group size (*Snijders et al., 2021*). Such experiments are extremely challenging to conduct, particularly in large vertebrates (*Krause and Ruxton, 2002*). A commonly used experimental approach is removing single individuals from already established groups. However, without experimentally establishing groups, biases between groups can persist, and removing individuals can lead to social upheaval resulting in variable and inaccurate estimates of reproductive success (*Dickinson and Hatchwell, 2004*; *Downing et al., 2020*).

To overcome these challenges, we studied cooperative breeding ostriches, *Struthio camelus*. Ostriches breed in groups that can range from pairs to large multi-male, multi-female groups (*Figure 1C*; *Bertram, 1992*; *Franz Sauer and Sauer, 1966*). The number of males and females within groups can be highly variable, but the distribution of group sizes is reported to be relatively stable across years (*Bertram, 1992*). Observations of recognisable individuals in the wild also suggest that males and females occupy the same areas over successive breeding seasons (*Bertram, 1992*). Groups are usually composed of unrelated individuals that all attempt to breed in a communal nest (*Kimwele and Graves, 2003*). Males and females cooperate over the incubation of eggs that lasts for approximately 42 days, representing a major part of parental care (*Kimwele and Graves, 2003*; *Magige et al., 2009*). During this period, eggs must be continually incubated and protected, exposing adults to the risks of heat exhaustion and predation (*Bertram, 1992*; *Magige et al., 2008*; *Franz Sauer and Sauer, 1966*).

We quantified the distribution of breeding group sizes in a wild population in the Karoo National Park (KNP), and compared it to the benefits of cooperative breeding in experimentally established groups in captivity ($n_{groups}$ = 96, $n_{individuals}$ = 273) to test two hypotheses: (i) male and female reproductive success is maximised across a range of group sizes due to the counteracting effects of cooperation and competition, and (ii) differences in the costs of competition and benefits of cooperation between males and females generate sex-specific reproductive payoffs from being in groups of different size. At the start of each breeding season in May, groups were experimentally established by placing different numbers of males and females in large enclosures in the Klein Karoo, South Africa. Groups consisted of one or three males and one, three, four, or six females, in accordance with the core range of group sizes seen in the wild (*Figure 1C*, *Figure 1—figure supplement 1*. *Supplementary file 1a*). All males were unrelated to females. A few same sex individuals were related, but this did not influence our results (Materials and methods section 'Supplementary analyses'). To separate the costs of sexual competition from the benefits of cooperation in groups of different size, we measured reproductive success both with and without opportunities for cooperative incubation. Opportunities for cooperative incubation were restricted by experimentally removing eggs from their nests and hatching them in artificial incubators for the first 5 months of each breeding season (*Figure 1D–E*). For the remaining 2 months of each breeding season, eggs were left in nests to allow cooperative incubation. This enabled us to estimate the mean reproductive payoffs for males and females in each group with and without opportunities for cooperative incubation (mean reproductive success = total number of chicks/number of same sex individuals in the group; data on the number of eggs produced are presented in the supplementary material: *Figure 2—figure supplement 1*, *Supplementary file 1b and c*). Parentage analysis of 3227 offspring verified that our measures of mean reproductive success accurately reflect genetic measures of individual reproductive success (*Figure 2—figure supplement 2*; Materials and methods sections 'Measuring reproductive success' and 'Genetic parentage analysis').

## Results

Groups of wild ostriches in the KNP were highly variable in size. Groups consisted of one to twelve individuals of the same sex, most often containing one to six females and one to three males (*Figure 1*; *Figure 1—figure supplement 1*; *Supplementary file 1a*). The composition of groups we observed was also similar to that reported in East African populations (*Bertram, 1992*; *Kimwele and Graves, 2003*; *Magige et al., 2009*). This shows that local variation in group sizes is widespread across the geographical distribution of ostriches, and was evident in wild populations close to the experimental study site exposed to similar climatic conditions (KNP is 170 km from the experimental population).

### Sexual competition regulates the optimal group size for males

In experimental groups, male reproductive success increased with the number of females and decreased with the number of males during the period when cooperation over incubation was prevented (*Figure 2A*; number of females β (credible interval: CI)=0.54 (0.26, 0.79), pMCMC = 0.001; males 3 vs. 1 β(CI)=−1.24 (−1.7, −0.86), pMCMC = 0.001; *Supplementary file 1d*). Allowing for cooperation over incubation reduced the effect of competition on male reproductive success, particularly in groups with fewer females (*Figure 2A–B*; number of females*number of males β(CI)=−2.12 (−4.05, −0.12), pMCMC = 0.008; *Supplementary file 1d*), and increased the benefits of having more females for single males (*Figure 2A–B*; number of females care vs. no care β(CI)=−1.74 (−4.1, −0.22), pMCMC = 0.008; *Supplementary file 1d*). However, this did not change the optimal group size for males: males produced most offspring when they were in groups on their own with four or more females, regardless of the benefits of cooperative incubation.

### Cooperation results in multiple group size optima for females

Female reproductive success was not related to the number of males and females in groups when incubation was prevented (*Figure 2C*; males 3 vs. 1 β(CI)=−0.1 (−0.39, 0.35), pMCMC = 0.856; number of females β(CI)=−0.1 (−0.28, 0.04), pMCMC = 0.13; *Supplementary file 1e*). However, when there were opportunities for cooperative incubation, female reproductive success was strongly dependent on both the number of males and females in groups (*Figure 2D*; *Supplementary file 1e*). In groups with single males, the number of offspring females produced increased linearly with the number of females (*Figure 2D*; number of females β(CI)=0.55 (0.04, 1.74), pMCMC = 0.008; *Supplementary file 1e*). In contrast, in groups with three males, females produced most offspring when on their own and when in groups with six females (*Figure 2D*; males 3 vs. 1: number of females$^2$ β(CI)=1.01 (0.2, 1.68), pMCMC = 0.004; *Supplementary file 1e*). Female reproductive success was therefore comparable across multiple group compositions due to the benefits of cooperating over incubation with other male and female group members (*Figure 2*; *Supplementary file 1e*).

### Cooperative care in larger groups increases hatching success and lightens workloads

We investigated how cooperative incubation influenced the group size optima for males and females by observing their behaviour. Incubation was often shared by both males and females and the total time eggs were incubated increased with the number of males and females in groups (*Figure 3A*; males 3 vs. 1 β(CI)=1.27 (0.29, 2.06), pMCMC = 0.006; number of females β(CI)=0.94 (0.56, 1.34), pMCMC = 0.001; *Supplementary file 1f*). In turn, hatching success was higher in groups where nests were incubated for a greater proportion of time per day (*Figure 3B*; β(CI)=0.35 (0.03, 0.7), pMCMC = 0.034; *Supplementary file 1g*). Individuals did not, however, spend more time incubating in larger groups *Supplementary file 1*. Males spent less time incubating when other males were in the group, even though the nests were incubated for a greater proportion of time (*Figure 3D*; males 3 vs. 1 β(CI)=−3.07 (−5.28, −0.76), pMCMC = 0.004; *Supplementary file 1h*), while the proportion of time females spent incubating was largely independent of group size (*Figure 3C*; *Supplementary file 1i*). The advantage of being in larger groups therefore appears to be explained by cooperation over incubation spreading the load of parental care and increasing hatching success.

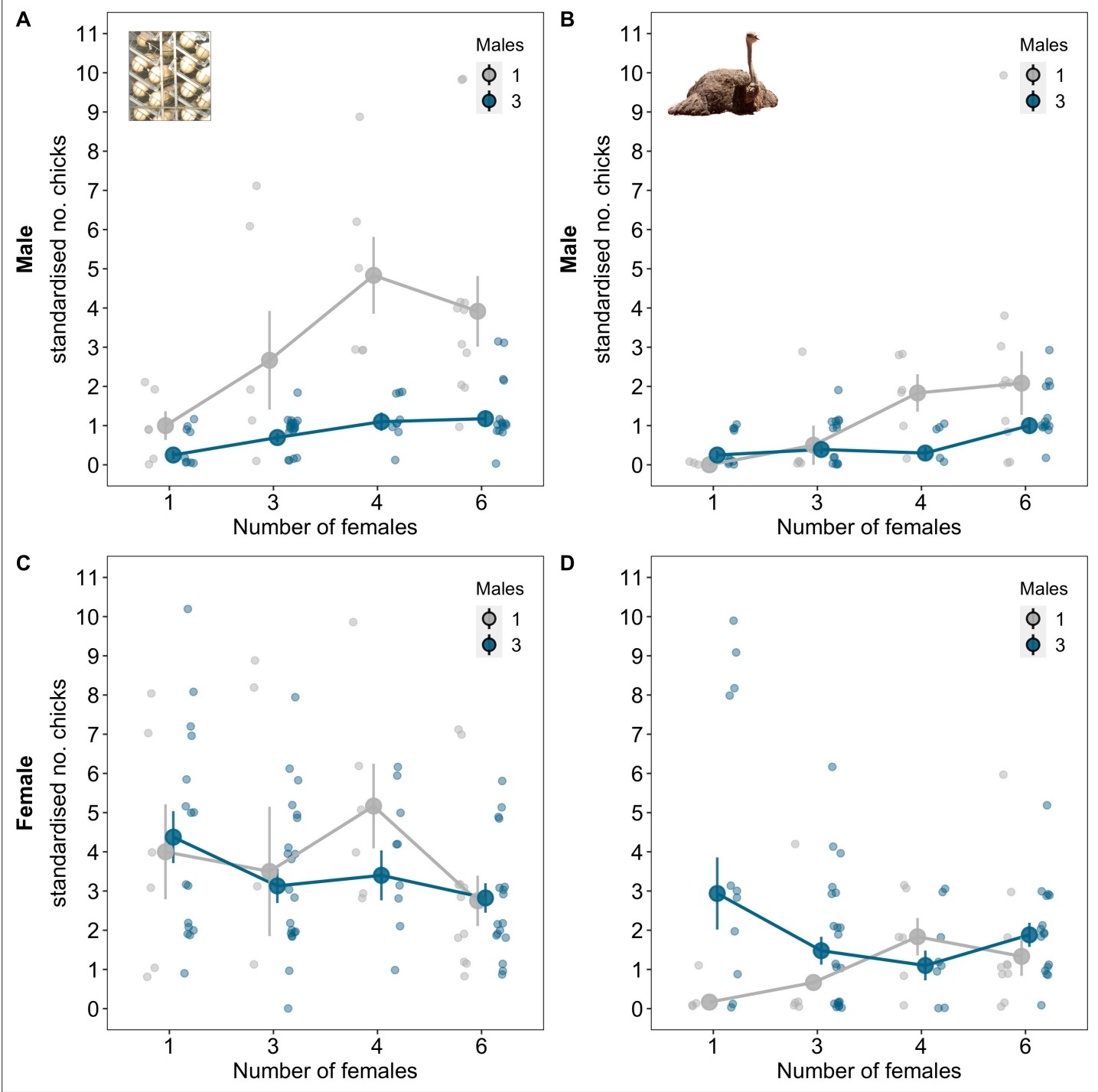

**Figure 2.** Group size and opportunities for cooperation over incubation influence male and female reproductive success. Reproductive success was measured as the number of chicks produced per individual per clutch for each reproductive stage (artificial vs. natural incubation; see Materials and methods for details). (**A**) The number of chicks males sired decreased with the number of males in the group and increased with the number of females. (**B**) Opportunities for cooperative incubation reduced the effects of male competition in groups with few females and magnified the effect of the number of females in groups without male competition (*Supplementary file 1d*). (**C**) When opportunities for cooperative incubation were removed the number of chicks females produced was independent of the number of males and females in groups. (**D**) When there were opportunities for cooperative incubation, the number of chicks females produced was dependent on both the number of males and females in groups. Means ± SE are plotted. Full details of sample sizes are presented in *Supplementary file 1r*.

The online version of this article includes the following figure supplement(s) for figure 2:

*Figure 2 continued on next page*

*Figure 2 continued*

**Figure supplement 1.** The effect of group size and opportunities for cooperation over incubation on the number of eggs produced per individual.

**Figure supplement 2.** The correspondence between measurements of reproductive success for males and females measured with and without genetic methods.

**Figure supplement 3.** The total reproductive output of groups in relation to group size and offspring care.

## Sexual conflict over the timing of mating and incubation in intermediate sized groups

Limited opportunities for cooperation over incubation may constrain the reproductive success of individuals in small groups (*Figures 2 and 3*). However, this does not explain why reproductive success was low in groups of intermediate size (~3 males and ~3 females). In some groups, males attempted to copulate with incubating females that resulted in females leaving nests and eggs being displaced and broken. We investigated whether such a lack of coordination over mating and incubation explained the reduction in the reproductive success of individuals in intermediate sized groups.

The number of interruptions to incubation increased with the number of males in groups (*Figure 4A*; males 3 vs. 1 (CI)=1.5 (0.5, 2.73), pMCMC = 0.002; *Supplementary file 1j*). This was most pronounced in groups with intermediate numbers of females (*Figure 4A*; number of females$^2$ β(CI)=−0.67 (−1.13, −0.16), pMCMC = 0.018; *Supplementary file 1j*). In contrast, in groups with the lowest and highest numbers of females, interruptions to incubation were relatively rare (*Figure 4A*; *Supplementary file 1j*). Interruptions to incubation were associated with a mismatch in the amount of time males and females spent incubating, but only in groups where there was male competition (*Figure 4B*; differences in incubation: 3 males β(CI)=−1.43 (−2.36, −0.26), pMCMC = 0.024, 1 male β (CI) = 0.18 (−1.4, 1.33), pMCMC = 0.994; *Supplementary file 1k*). As a result, interruptions were more frequent in groups with multiple males where females incubated for longer than males, but not where males invested more time than females in incubation (*Figure 4B*).

Disparities in incubation between males and females influenced the probability of egg breakages. Eggs were more frequently destroyed when females invested more time in incubation than males (*Figure 4C*; differences in incubation in groups with 3 males: β(CI)=−0.64 (−1.23, −0.06), pMCMC = 0.026; *Supplementary file 1l*), which reduced hatching success (*Figure 4D*; β(CI)=−0.82 (−1.2, −0.58), pMCMC = 0.001; *Supplementary file 1m*). A discrepancy between males and females in the timing of mating and incubation therefore appears to explain the lower reproductive success of individuals in groups with intermediate numbers of males and females.

## Discussion

Our experiments show that the counteracting forces of cooperation and competition on patterns of male and female reproductive success can help explain group size variation in cooperative breeding societies. While the change in the costs and benefits of increasing numbers of males and females resulted in a clear optimal group size for male ostriches, this was not the case for females. The benefits of cooperative incubation and the costs of sexual conflict led to female reproductive success being maximised across multiple group sizes. Although the importance of competition and cooperation for group living species has long been recognised (*Alexander, 1974*; *Williams, 1966*), our results demonstrate that differences in their relative strengths in males and females can select for different group sizes and compositions.

In species where group members are typically unrelated, ongoing sexual conflict over parental care and parentage is generally thought to explain variation in breeding systems, both within and between species (*Davies, 1989*; *Lessells, 2012*; *Trivers, 1972*; *Wong et al., 2012*). For example, in dunnocks, *Prunella modularis*, and alpine accentors, *Prunella collaris*, female reproductive success is increased by the number of males providing care, but male reproductive success declines due to more intense sexual competition (*Davies et al., 1995*; *Davies, 1992*; *Davies, 1985*; *Hartley and Davies, 1994*; *Santos and Nakagawa, 2013*). Consequently, polyandrous breeding systems are thought to arise where females have the upper hand, and polygynous breeding systems where males have more control (*Davies, 1989*).

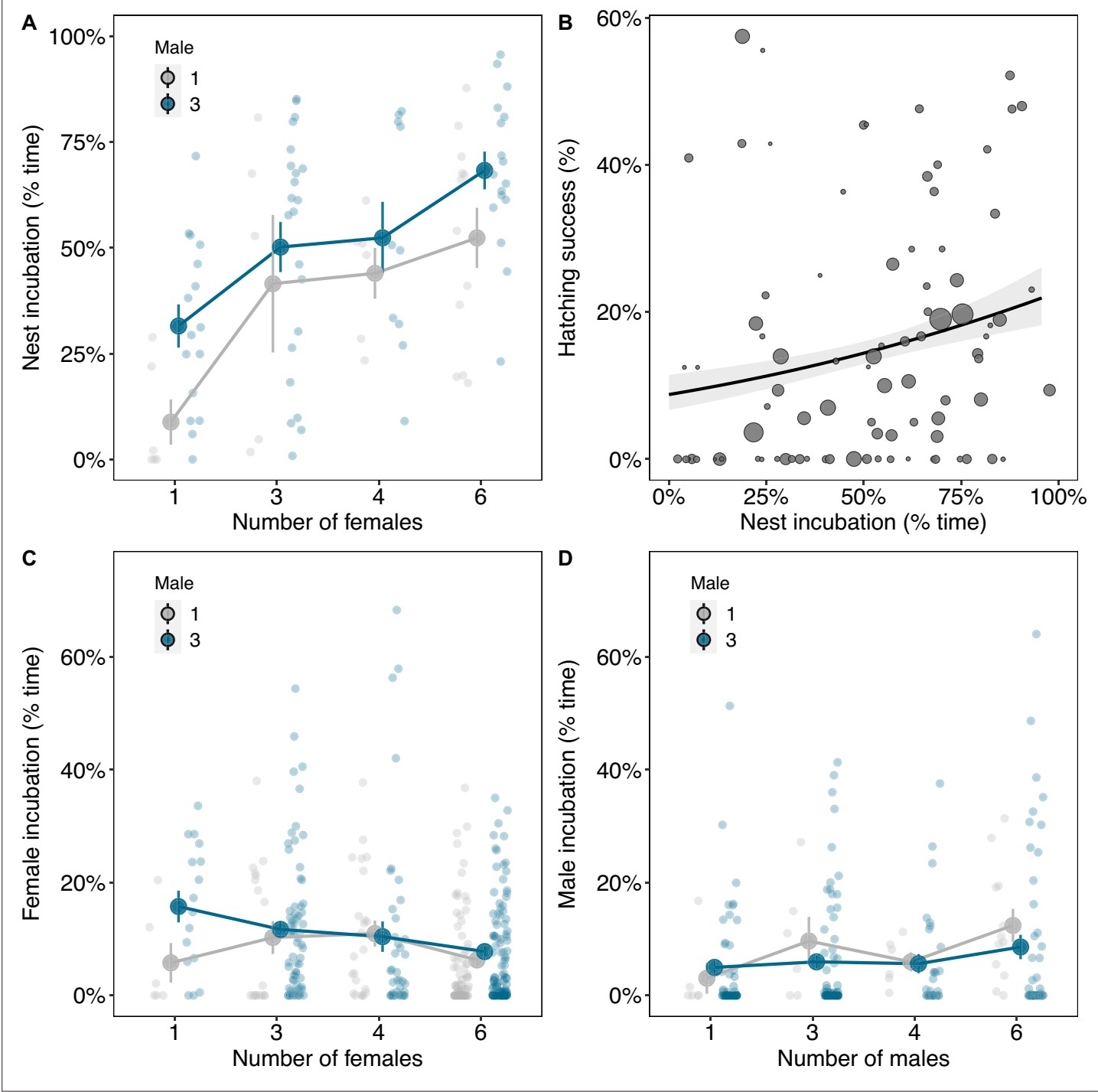

**Figure 3.** The benefits of cooperative parental care in relation to group size. (**A**) The amount of time nests were incubated was higher in groups with more males and females. (**B**) Hatching success increased with the amount of time nests were incubated. Regression line from a binomial generalised linear model (GLM) with 95% confidence intervals is shown and the size of the points represents the number of eggs laid by groups. (**C**) The amount of time females spent incubating decreased with the number of females in groups, although not significantly. (**D**) Males spent less time incubating in groups with three males compared to when they were on their own. Means ± SE are plotted in A, C, and D. Full details of sample sizes are presented in *Supplementary file 1r*.

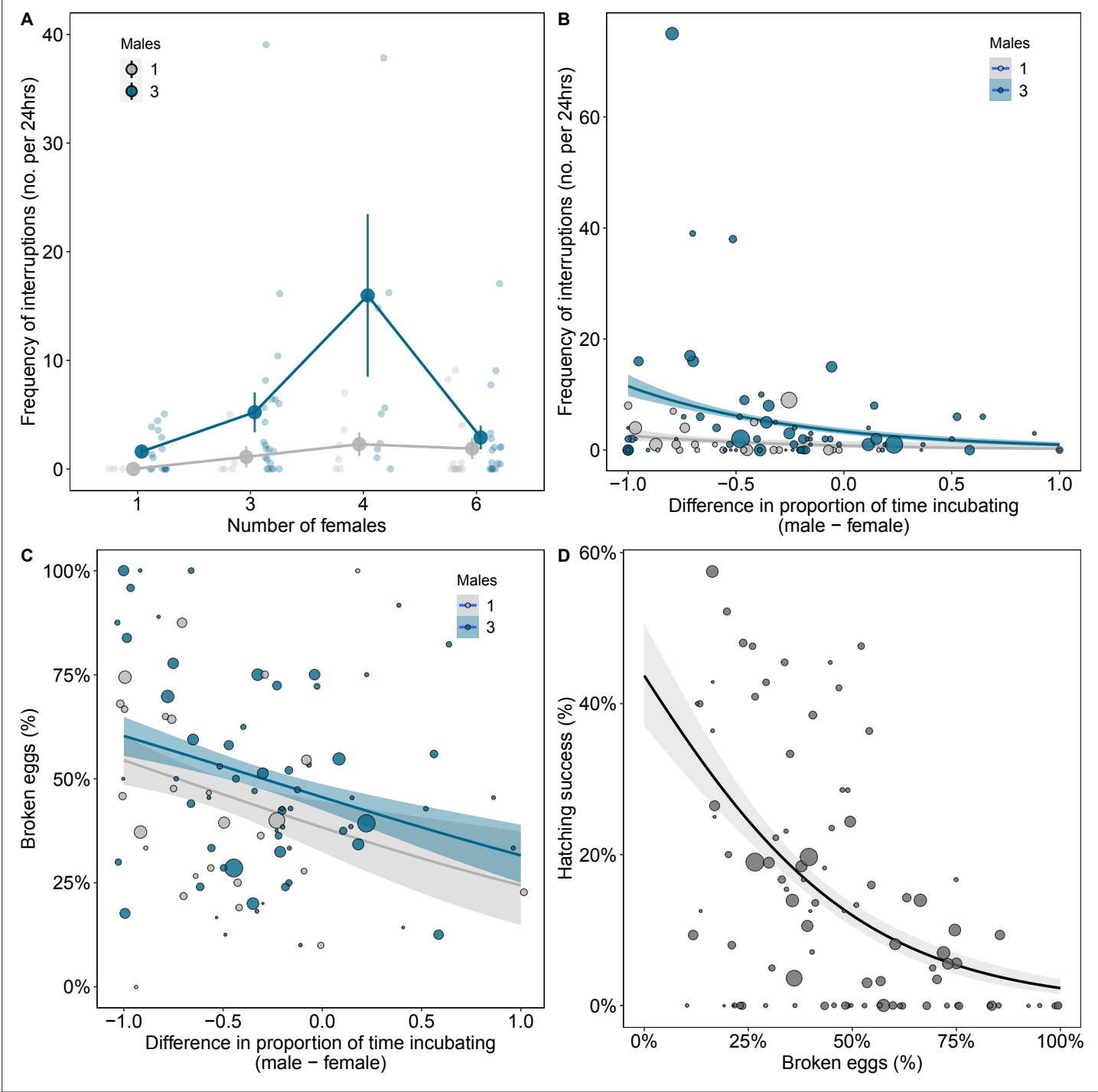

**Figure 4.** Coordination over reproduction changes with group composition. (**A**) The frequency of interruptions per 24 hr period of incubation increased with the numbers of males in groups, especially when there were intermediate numbers of females. Means ± SE are plotted. Full details of sample sizes are presented in ***Supplementary file 1r***. (**B**) Interruptions to incubations were more frequent in groups with three males when females spent more time incubating than males. (**C**) More eggs were broken in groups when disparities in the time males and females spent incubating were greater, which decreased hatching success (**D**). Regression lines from generalised linear models (GLMs) (B=Poisson; C and D=binomial) with 95% confidence intervals are presented for graphical purposes. The size of the points in B, C, and D represents the number of eggs laid by groups.

Sexual conflict also appears to influence the types of social groups observed in ostriches. The pursuit of reproductive opportunities by males beyond the point where females engage in parental care appears to lead to a 'sexual tragedy of the commons', causing the demise of groups with intermediate numbers of males and females (*Le Galliard et al., 2005*; *Hardin, 1968*; *Rankin et al., 2011*). Conflicts over the timing of reproduction between the sexes have been found to influence reproductive success in other species (*Holland and Rice, 1998*; *Løvlie and Pizzari, 2007*). Our results show that such conflicts may also shape the composition of cooperative breeding groups. However, only a relative small amount of variation in the reproductive payoffs of being in groups of different size appeared to be due to sexual conflict (*Figure 2*). Instead, differences in the benefits of cooperation and the costs of sexual competition for males and females seemed to more important for explaining patterns of reproductive success across group sizes.

Our results show that monitoring only the outcome of breeding events makes it difficult to estimate the relative contributions of mate choice, sexual competition and cooperation to selection for group living. For example, without manipulating the need for cooperation it would have been difficult to ascertain if the elevated success of single females in groups with three males was because of differences in mate choice opportunities, or because of the extra parental care provided by males. Given that the reproductive success of single females did not vary with the number of males when opportunities for cooperation were restricted, it appears that female reproductive success was increased by paternal care and not by mate choice. This highlights the importance of experiments in identifying the sources of selection shaping the social organisation of cooperative breeding animals.

Under natural conditions, group sizes are likely to depend on a variety factors, such as predator defence and food availability (*Alexander, 1974*; *Koenig and Dickinson, 2016*; *Rubenstein and Abbot, 2017*; *Taborsky et al., 2021*). Evidence from mammals, fish, birds, and arthropods shows that the size of cooperative groups often varies with ecological conditions (*Bertram, 1980*; *Bourke, 1999*; *Clutton-Brock, 2021*; *Koenig, 1981*; *Yip et al., 2008*; *Zöttl et al., 2013*). In experiments such as ours, it is difficult to examine the effect of ecological pressures on the benefits of breeding in different sized groups. However, the results of experimental studies that control for ecological conditions can be used as a benchmark to compare data from wild populations (*Figure 1—figure supplement 1*). In this way, it is possible to more accurately assess the contributions of ecological factors and social interactions to the costs and benefits of living in cooperative breeding groups.

Comparing the group sizes we observed in the KNP with our experimental data shows a general trend for groups in the wild to be smaller than those that maximise reproductive success in captivity (*Figure 1—figure supplement 1*). For example, pairs are frequently observed in the wild (see also *Franz Sauer and Sauer, 1966* and *Bertram, 1992*), but had low reproductive success in our experiment. There are many possible reasons for this, such as larger groups being more visible to nest predators in the wild (*Bertram, 1992*; *Bertram, 1980*) and food being more limited compared to captivity. It is also possible that observed individuals were in the process of forming groups, given nests are difficult to find, or were constrained in their choice of groups to join (*Sibly, 1983*). Nevertheless, data from experimental manipulations provides a crucial platform to idenitfying patterns of sociality in the wild that require further explanation.

In summary, our results demonstrate that variation in cooperative breeding groups arises independently of ecological conditions, breeder quality, and relatedness, the main factors often invoked to explain variation in cooperative breeding groups (*Koenig, 1981*; *Rubenstein and Abbot, 2017*; see also *Casari and Tagliapietra, 2018*). The counteracting forces of competition and cooperation can generate divergent reproductive interests between males and females and increase the range of social groups where reproductive success is maximised. Establishing how social interactions influence selection on group living is an important step in interpreting patterns of sociality in nature.

## Materials and methods
### Study population

This study was conducted on two populations. Natural variation in group composition was examined in a wild population of ostriches in KNP, South Africa (32°19'49.27"S, 22°29'59.99"E) during the middle of their natural breeding season (*Franz Sauer and Sauer, 1966*) in 2014 (8–9 November) and 2018 (17–19 November). The experiments manipulating group size were conducted on a captive

population of ostriches kept at Oudtshoorn Research Farm, South Africa (33° 38' 21.5"S, 22° 15' 17.4"E) from 2012 to 2018 during the months of May to December. The founders of the captive population used in this study originate from different farms across South Africa. They are genetically similar to Southern African populations (*Davids et al., 2012*), and are commonly referred to as South African Blacks (*Struthio camelus* var. *domesticus*).

## Natural variation in group composition

Marking and tracking of wild individuals is difficult so little is known about group stability, but the distribution of group size has been reported to be relatively stable over time (*Bertram, 1992*). Natural variation in the composition of breeding groups (group size, number of males, and number of females) was examined using published literature (*Bertram, 1992*; *Kimwele and Graves, 2003*; *Magige et al., 2009*), and directly estimated by conducting transects along the roads of the southeastern part of the KNP. Each transect was carried out two to three times. Ostriches were typically observed in clearly defined groups, judged by their coordinated movement and close proximity to each other (<~100 m). In a few instances, individuals were separated by more than 100 m. In these situations they were observed until it was clear whether they were part of a group or were moving separately. Whether individuals were sexually mature (immature females = no or very few white wing feathers; immature males = mix of brown and black body plumage) was also recorded. Immature individuals were observed only twice; one group of three males and one female, approximately 2 years of age, and one group of seven individuals, approximately 1 year of age and therefore not possible to sex. *Figure 1C* includes the 2-year-old group, but not the 1-year-old group. We noted whether adult males observed in the KNP were in breeding condition by the colouration of their bill and tarsal scales, which becomes red during the breeding season (*Bertram, 1992*). Groups composed of males in breeding condition and adult females were assumed to be breeding.

## Experimental design

We experimentally manipulated the composition of 97 groups of breeding ostriches in captivity involving 280 adult ostriches (127 males and 153 females), over a 7-year period (10–16 groups per year determined by the availability of birds and enclosures). Groups were kept in fenced areas (range: 2400–70,600 m$^2$, median = 4700 m$^2$) at the Oudtshoorn Research Farm (*Cloete et al., 2008*) and were randomly allocated one or three males and one, three, four, or six females. Due to limitations in the number of birds accessible for our experiments, and other experiments being conducted on the same population, not all combinations of male and female group sizes were possible (see *Supplementary file 1r* for full details). The number of males and females in groups were varied independently, enabling sex differences in optimal group size to be estimated, and interactions between the number of each sex (sex ratio) to be accounted for. All individuals in the Oudtshoorn population were individually identifiable by coloured and numbered neck tags.

In seven groups, an individual was injured or died part way through the season and was replaced by a new individual of the same sex. In 12 groups, spread across years and group sizes, it was not possible to replace injured individuals. The number of individuals was consequently reduced (six groups = nine to eight, one group = seven to six, three groups = six to five, and one group = five to four; see *Supplementary file 1s* for details of replacements and removals). To avoid creating new group size treatments with low sample sizes, these groups were treated as part of their intended group size treatments in the analyses. In one case the injured individual was the only male in the group and we therefore only included data from the point when the replacement male was introduced (see *Supplementary file 1s*). One group was excluded from all analyses as the injured individual was the only female in the group. Our final sample size was therefore 96 groups (*Supplementary file 1r*).

The breeding season was from May to December each year. During the first ~5 months of the season, eggs were collected to measure reproductive success independently from the effects of incubation behaviour. During the last ~2 months, eggs were left in nests and incubation behaviour was monitored to examine patterns of reproductive success when individuals had to care for the brood. Reproductive success was measured as the number of eggs and number of chicks produced by groups. During the breeding season ostriches received a balanced ostrich breeder diet (90–120 g protein, 7.5–10.5 MJ metabolisable energy, 26 g calcium, and 6 g phosphorus per kg feed) and ad libitum water.

## Measuring reproductive success

### Reproductive success when opportunities for cooperative incubation were removed

To measure reproductive success independently of incubation behaviour, eggs were collected from nests twice a day and artificially incubated. Eggs were marked according to the time of day, date and group of origin, and placed under UV lights for 20 min for disinfection. As eggs were incubated in batches starting each week. Eggs were stored prior to incubation for 1–6 days under conditions known to maintain hatching success (**Brand et al., 2008**): Eggs were kept on turning trays (two daily 180° rotations) in a cold room (17°C) with relative humidity between 80% and 90%. After storage, eggs were transferred to artificial incubators set at 36.2°C with a relative humidity of 24%. Eggs in the incubator were automatically turned 60° around their long axis every hour. Eggs were inspected daily for signs of pipping from day 39 of incubation when they were moved to hatchers. The incubation period in ostriches is ~42 days. We were interested in the average reproductive returns for individuals in groups of different size, irrespective of between individual variation in reproductive success within groups. To do this, we estimated individual reproductive success as the number of eggs and chicks produced by groups, divided by the total number of same sex individuals within groups. Therefore, there was one measure of reproductive success for females and another for males in each group. To verify that our measure of average reproductive success per individual accurately reflected the genetic measures of reproductive success, we genotyped 3227 chicks hatched out in artificial incubators (Materials and methods section 'Genetic parentage analysis'). There was an extremely strong correlation between the average number of chicks produced per individual and genetic measures of the number of chicks males (R=0.96) and females (R=0.95; **Figure 2—figure supplement 2**) produced.

### Reproductive success when there were opportunities for cooperative incubation

Nests were checked daily and new eggs were marked with the date and an egg identification number. The absence and presence of previously laid eggs was recorded to track the fate of each egg. During the period when eggs were left in the nests, the incubation behaviour of individuals was monitored by conducting ~3 hr observations at least three times a week using binoculars (10×40) and a telescope (12–36 × 50). Groups were monitored when opportunities for cooperative incubation were removed, but incubation behaviour was only recorded when eggs were left in nests. The observer sat camouflaged in a 10 m tall observation tower in the middle of the field site. Groups were observed for an average of 60.54±1.38 hr (mean ± SE) per year spread over ~2-month period when eggs were left with groups. The identity of each incubating individual, as well as the start and end of incubation, were recorded. When incubation was interrupted by a copulation attempt, the time of the interruption and the identity of the individuals involved in the interruption were recorded. The consequences of interruptions varied in severity from individuals returning to nests within seconds to individuals ceasing incubation for that observation period. To avoid including minor disturbances in our measure of the number of interruptions, we only included interruption events that resulted in the incubating individual not returning to the nest within 1 min.

In the first 3 years (2012–2014), hatching success was measured by allowing groups to naturally incubate eggs to completion. Hatching success was defined by the proportion of eggs that hatched out of the total number of eggs produced during the period when eggs were left with groups. If no eggs were observed hatching in groups after 50 days of incubation, they were removed. From 2015 onwards, changes in legislation to reduce the spread of avian flu meant that contact between adults and chicks had to be minimised. Consequently, eggs were removed from nests just before hatching (~40 days after the onset of incubation) and placed in artificial incubators to determine hatching success. Individual reproductive success was estimated in the same way as when eggs were artificially incubated: the number of eggs and chicks produced by groups divided by the total number of same sex individuals within groups.

### Standardising reproductive success

To be able to compare patterns of reproductive success between periods when eggs were artificially and naturally incubated, and across different years with slightly different breeding season lengths,

we standardised the number of eggs and chicks produced by individuals. First, the number of eggs individuals produced was divided by the number of days that reproduction was monitored to get a measure of reproductive output per day. Second, measures of individual reproductive output per day were divided by the maximum egg output for that sex, for that stage of the experiment (no care vs. care). This was important because hatching success was much higher when eggs were artificially incubated compared to when eggs were naturally incubation. This gave a maximum egg laying rate per day of one for both stages. However, females can biologically only produce one egg every 2 days. We therefore divided our standardised measures by two and multiplied it by the number of days it typically takes to form a clutch (20 days) to obtain a measure of the eggs laid per clutch. The same procedure was used to standardise the number of chicks individuals produced.

## Genetic parentage analysis

We collected blood and tissue samples from adult individuals in our experimental groups and their offspring (chicks and unhatched eggs). All procedures were approved by the Departmental Ethics Committee for Research on Animals (DECRA) of the Western Cape Department of Agriculture, reference no. AP/BR/O/SC14. During the initial 3 years of the study (2012–2014), the parentage of 1860 offspring samples were analysed using seven highly polymorphic microsatellites previously shown to assign parentage with high confidence in ostriches (**Bonato et al., 2009**). The number of samples analysed per group was chosen according to the number of females in the group, multiplied by 10 where possible, spread across the breeding season (range of samples per group = 3–72). Microsatellites were amplified using Phusion Blood Direct PCR Kit (Thermo Fisher Scientific) and those of similar length were differentiated using fluorescent primer tagging (HEX and FAM). After DNA amplification, the amplicons were separated by size using capillary electrophoresis. Microsatellite scoring was performed using Geneious 10.2.3 (https://www.geneious.com).

The most likely parent pair for every offspring was assigned using Cervus 3.0.7 (**Marshall et al., 1998**). Initially, candidate parents were restricted to adults in the group. Mislabelled eggs, chicks, and blood samples are, however, possible. Consequently, an additional analysis was performed for offspring that were unassigned after our initial analysis where all adults present in experimental groups in that year were regarded as candidate parents.

Out of the 1860 offspring selected for parentage analysis, 1736 (93.3%) were assigned parentage with strict confidence (trio confidence ≥95%), 1 (0.05%) was assigned parentage with relaxed confidence (trio confidence >80 but<95%), 120 (6.4%) were assigned parentage with low confidence (trio confidence <80%), and 3 (0.2%) remained unassigned.

In the final 4 years of the study (2015–2018), parentage was analysed using single-nucleotide polymorphisms (SNPs) of up to 37 offspring per group (we aimed to sample 20 eggs during the artificially incubation phase, and 20 from when eggs were left in nests. Total samples = 1377, range of samples per group = 5–37). DNA was extracted from ~25 ng dry blood or tissue stored in 95% ethanol at –20°C. The samples were placed in 100 µl of lysis buffer (0.1 M Tris, 0.005 M EDTA, 0.2% SDS, 0.2 M NaCl, pH 8.5) and 1.5 µl proteinase K (~20 mg/ml) and vortexed for ~1 min. Samples were incubated for 3 hr at 56°C, with an additional ~1 min vortex after 1 hr of incubation. After the incubation, samples were centrifuged for 10 min at 11,000× $g$. The supernatant was put into 10 µl of NaAc (3 M) ad 220 µl of ice-cold (–20°C) 99% ethanol was added to precipitate the DNA. Samples were centrifuged for 10 min at 11,000 rpm. The precipitation procedure was repeated with 100 µl ice-cold 70% ethanol. Finally, the DNA was dried in a vacuum centrifuge for 10–15 min and dissolved in 50 µl ×1 TE buffer. The genotyping of SNPs from extracted DNA was done using SNPtypeTM assays for locus specific PCR amplification (a so-called specific target reaction or STA) followed by allele specific PCR reaction targeting 96 markers in a 96.96 Dynamic ArrayTM IFC developed by Fluidigm inc (San Fransisco, USA). The genotype data was collected by allele specific imaging of fluorescent signals from each PCR-reaction using EP1TM reader (Fluidigm Inc). Genotyping was done using the instructions from Fluidigm with slight modifications to the STA for increased amplification. The modifications included the use of 2 µl template DNA, 28 PCR cycles, and dilution of the STA product 1:8 times prior to allele specific PCR. The SNP Genotyping Analysis 4.5.1 software (Fluidigm) was used for genotype calling. We used the SNPtype normalization method and a confidence threshold of 0.65 and manually validated all scatter plots for proper clustering and separation of the different genotypes. Samples were analysed in two replicates

and consensus genotypes constructed solely from markers providing identical genotypes at both replicates.

Markers were identified using whole genome sequencing data from five males and five females of three different ostrich populations (*S. c. var. domesticus*, *S. c. massaicus,* and *S. c. australis*) kept at the Oudtshoorn Research Farm (total individuals = 30). We used pedMine version 1.0.0 (*Douglas and Sandefur, 2008*) to identify the individuals with the most distant links in the population pedigree (*Schou et al., 2022*), allowing the maximum amount of genetic diversity in populations to be sampled. Samples were sequenced at Science for Life Laboratory, the National Genomics Infrastructure, using Illumina HiSeq 2500, following the manufacturer's protocol. Sequencing was done on six different flow cells, three each in 2015 (2×126 bp) and 2016 (2×150 bp).

Variant calling was done using a best practice workflow for variant calling developed at the Broad Institute (*Depristo et al., 2011*). Briefly, reads from separate batch runs were adapter-trimmed with Cutadapt (*Martin, 2011*) version 2.3 with options `--trim-n -q10,10 -m 80` and then mapped to the reference genome with bwa (*Li and Durbin, 2009*) version 0.7.17-r1188. We mapped 201–453 million reads for samples with mapping frequencies in the range of 93.4–96.1%. The average depth of coverage was between 22 and 50.

Picard (http://broadinstitute.github.io/picard/; *Picard Toolkit, 2019*) version 2.20.0 was used to sort the mapped reads, add read group information, and then merge batch runs to sample level bam file. GATK (*Depristo et al., 2011*) version 4.0.9.0 was used to identify and realign indels, where after duplicates were marked with Picard MarkDuplicates.

A first round of raw variant calling was performed with GATK HaplotypeCaller, freebayes (*Garrison and Marth, 2012*) version 1.2.0 and bcftools (*Li, 2011*) version 1.9. GATK HaplotypeCaller was run in GVCF mode (option -ERC GVCF). Variant calling was first run on each sample separately, followed by genotyping with GATK GenotypeGVCFs with options –max-genotype-count 1000. freebayes was run jointly on all samples and sex, so all regions were treated as having ploidy 2, with options `--min-mapping-quality 5 --min-base-quality 5 --use-mapping-quality --use-best-n-alleles` 4. As freebayes may perform poorly in regions with excessive coverage, regions of excessive coverage were excluded during variant calling. Finally, bcftools was run jointly on all samples and sex with unadjusted ploidy settings. The three call sets were merged and intersected to generate a high-quality call set of known sites. Briefly, indels and private SNPs were removed from all call sets. Call sets were intersected and only SNPs present in at least two call sets were kept.

The 96 autosomal SNPs were identified using criteria similar to those of *Andrews et al., 2018*. In addition to the variant quality filtering already imposed, we removed genotypes not called in all 30 individuals with a genotype quality of ≥30 or not present on scaffolds with a minimum size of 1 Mb. We used PLINK 1.90 (*Purcell et al., 2007*) to remove loci not in Hardy-Weinberg equilibrium (p<0.05) using the mid-p adjustment. We extracted SNPs with a minor allele frequency (MAF) ≥ 0.45. These variants were pruned such that no pair of variants within a 5000 kb window had a squared allele count correlation ($r^2$) above 0.05. As this pruning technique does not remove closely adjacent variants if they are not in linkage, we randomly removed variants such that there was maximum of one variant per 500 kb window to give enough candidates. The Fluidigm protocol is sensitive to adjacent variants (<30 bp) and high GC content. Candidate variants that had adjacent variants and sequences with GC content above 65% were therefore removed.

Parentage was analysed using the function *MLEped in* the R-package MasterBayes vs. 2.57 (*Hadfield et al., 2006*). The probability of an allele being miss scored was set to the default value of 0.005. We initially refrained from defining any prior requirements of the group of offspring and adults, given eggs and samples can be mislabeled. Blood samples were not obtained for two adult males and two adult females. These unsampled adults were included in the population when estimating parentage.

Parentage was assigned with 95% confidence for 1336 offspring (97.0% of the sampled offspring). For five offspring (0.36%), the assigned parents were in two different groups. As this is not possible, we reclassified these as unassigned offspring. For 16 offspring (1.14%), the assigned sire and dam were within the same group, but different to that recorded for the offspring. We therefore re-assigned those offspring to the group with the matching parents as it is possible that groups are incorrectly recorded when eggs are collected. Given the low rate of misassignments and the confidence by which we could assign them, we re-ran the analyses with several adjustments to assign parentage to

the remaining 41 unassigned offspring: unsampled adults were omitted and candidate parents were restricted to the same group as offspring. This provided full parentage with 95% confidence for additional 18 offspring. Of the remaining 23 unassigned offspring, 13 were from groups with one of the four unsampled adults. We therefore ran separate models for each of these groups. This was done with Markov chain Monte Carlo (MCMC) simulations in the function MCMCped, as this allowed us to inspect the posterior distribution of parentage for each offspring. All 13 offspring were assigned full parentage with 95% confidence, resulting in a total of 1367 (99%) assigned offspring samples.

## Statistical analyses

### General approach

Data were analysed in R (*R Core Team, 2020*) using Bayesian linear mixed models (BLMM) with MCMC estimation in the package MCMCglmm version 2.29 (*Hadfield, 2010*). Default fixed effect priors were used (independent normal priors with zero mean and large variance ($10^{10}$)) and inverse gamma priors were used for random effects unless otherwise specified (V=1, nu = 0.002). Each analysis was run for 3e+06 iterations with a burn-in of 1e+06 and a thinning interval of 2000. Convergence was checked by running models three times and examining the overlap of traces, levels of autocorrelation, and calculating Gelman and Rubin's convergence diagnostic (potential scale reduction factors <1.1) (*Brooks and Gelman, 1998*).

Parameter estimates (β) for fixed effects were calculated using posterior modes and are reported from full models with all terms of the same order and lower. For example, all main effect estimates are from models where all other main effects are included, all estimates of two-way interactions are from models that included all two-way interactions and main effects, and so forth. Quadratic effects are reported from models that included main effects and effects of the same order (other quadratic effects and two-way interactions). The length of time that groups were monitored was accounted for by including it as a fixed effect. All continuous explanatory variables were z transformed using the 'scale' function in R. Explanatory variables that were proportions were logit transformed using the 'logit' function in R and count variables were log transformed. Curvilinear effects of continuous explanatory variables were modelled using the quadratics of the z transformed values computed before running the models.

Fixed effects were considered significant when 95% credible intervals (CIs) did not overlap with 0 and pMCMC were less than 0.05 (pMCMC = proportion of iterations above or below a test value correcting for the finite sample size of posterior samples). By default MCMCglmm reports parameter estimates for fixed factors as differences from the global intercept. This does not allow absolute estimates and 95% CIs for all factor levels to be estimated, or custom hypothesis tests of differences between factor levels. Consequently, we removed the global intercept from all models and present absolute estimates for factor levels. Differences between factor levels were estimated by subtracting the posterior samples of one level from the second level and calculating the posterior mode, 95% CI, and pMCMC.

Random effects were used to model the non-independence of data arising from multiple data points per individual, per group, per enclosure, and per year. Random effect estimates presented in tables are from models that included the highest order of fixed effect terms. To estimate the magnitude of random effects, we calculated the percentage of the total random effect variance explained by each random term on the expected data scale (I2): ($V_i/V_{total}$*100) (*Devillemereuil et al., 2016*). To obtain estimates of I2 on the expected scale from binomial models, the distribution variance for the logit link function was included in the denominator: ($V_i/V_{total} + \pi^{3/2}$)*100.

## Specific analyses

### Testing how group size and the need for parental care influences male and female reproductive success

The effect of group composition on the standardised number of eggs individuals produced was modelled using a BLMM with a Poisson error distribution. Opportunities for incubation (two-level factor: no care vs. care), number of males (two-level factor: one vs. three), the number of females (continuous), and the time groups that were monitored (continuous) were entered as fixed effects, and year and enclosure were included as random effects. The effects of group composition on the number of eggs produced per adult with and without the need for incubation were estimated by

fitting two-way interactions between care and the number of males and females in groups. We also fitted three-way interactions between care, number of males and number of females in groups, as well as care, number of males and the quadratic number of females. Separate models were run for males and females (*Source code 1*: M1 and M2). The number of chicks males and females produced was modelled in exactly the same way (*Source code 1*: M3 and M4).

## Testing how the benefits of cooperative parental care vary with group size

The effect of group composition on the proportion of time that groups spent incubating nests was modelled using a BLMM with a binomial error distribution. The response variable was the number of observation minutes birds were sitting on nests relative to the number of observation minutes nests were exposed. This accounts for variation across years in observation effort. The number of males, the number of females, and the quadratic of number of females in groups were included as fixed effects and year and enclosure were fitted as random effects (*Source code 1*: M5). Two-way interactions between number of males and number of females, and between number of males and the quadratic of number of females were included. The effect of the proportion of time that nests were incubated on hatching success was modelled using a BLMM with a binomial error distribution of the number of eggs hatched by groups relative to the number of eggs that did not hatch. The same fixed and random effects were included as M5 with the addition of the proportion of time nests were incubated (*Source code 1*: M6).

Typically, groups only had one active nest, but in a few cases a second and a third nest were occasionally used. The amount of time groups incubated their nests was calculated by summing data across all nests (total time nests were incubated vs. total time nests were exposed). Data were summed across nests to facilitate comparisons with the egg and chick data, which were recorded at the level of the group (e.g. total number of eggs and chicks groups produced by each group), not at the level of each nest. To check if the number of nests groups used influenced the time nests were incubated and hatching success, we included the number of nests (continuous) as a fixed effect (*Source code 1*: M5 and M6). The number of nests did not have a significant effect in any of our analyses (*Supplementary file 1f and g*).

## Testing how individual investment in cooperative care varies with group size

The effect of group composition on the time individuals invested in incubation was modelled using a BLMM with a binomial error distribution. The response variable was the number of observation minutes an individual was sitting relative to the number of minutes it was not sitting, which accounts for variation in the amount of time individuals were observed. The number of males and the number of females in groups as well as the quadratic of number of females were included as fixed effects and year, enclosure, group, and individual identity were included as random effects. Two-way interactions between the number of males and the number of females, and between number of males and the quadratic of number of females were included. Separate models were run for males and females (*Source code 1*: M7 and M8). For this analysis only data on primary nests were included as attendance at secondary and tertiary nests was sporadic, and the presence of secondary and tertiary nests did not influence the total amount of time groups incubated their nests (*Supplementary file 1f and g*).

## Testing how male female coordination over incubation changes with group size

The effect of group size on coordination over incubation, measured as the number of male interruptions to female incubation, was modelled using a BLMM with a Poisson error distribution. The response variable was the total number of interruptions observed across all observations divided by the number of hours groups were observed (this was multiplied by 100 and rounded to whole numbers as MCMC-glmm requires count data to be whole numbers). The number of males, the number of females, and the quadratic of the number of females in groups were included as fixed effects, and year and enclosure were included as random effects (*Source code 1*: M9). Two-way interactions between number of males and number of females, and between number of males and the quadratic of number of females were included. We removed five groups where no incubation was observed given this removes the possibility for interruption. The effect of the disparity in the time males and females invested in incubation on the number of interruptions was modelled in the same way, but an extra fixed effect of the

difference in the proportion of time males and females spent incubating was included (*Source code 1*: M10).

## Testing how coordination over reproduction influences reproductive success

The effect of interruptions on the proportion of eggs broken in nests was modelled using a BLMM with a binomial error distribution. The response variable was the number of eggs broken relative to the number of eggs not broken. The difference in the proportion of time males and females spent incubating, the number of males in groups and their interaction were included as fixed effects, and the year and enclosure were included as random effects (*Source code 1*: M11). The impact of the broken eggs on the overall hatching success of groups was modelled using a BLMM with a binomial error distribution with the number of eggs hatched vs. the number of eggs that did not hatch as the response variable. The proportion of eggs that were broken was included as a fixed effect, and year and enclosure were included as random effects (*Source code 1*: M12).

## Acknowledgements

We are thankful to Tobias Uller, Philip Downing, Heikki Helanterä, Samuel Diaz-Munoz, Christian Rutz, Ralf Kurvers, and an anonymous reviewer for comments on the manuscript, to Bengt Hansson, Mikael Åkesson, Anna Danielsson, and Per Unneberg for advice and assistance with sequencing and parentage analysis, to all the staff at Oudtshoorn Research Farm for assistance with data collection and maintenance of the birds, and to the Western Cape Government for use of their resources. This research was funded by the Swedish Research Council (grant number 2017-03880), the Knut and Alice Wallenberg Foundation (Wallenberg Academy fellowship numbers 2013.0129 and 2018.0138), Carl Tryggers (grant numbers 12: 92 and 19: 71) to CKC and a stipend from the Carlsberg Foundation to MFS and by the Jörgen Lindström stipendium to JM. The maintenance and development of the ostrich population used in this study was funded by the Western Cape Department of Agriculture and supported by grants from the Western Cape Agricultural Research Trust (Grant number 0070/000VOLSTRUISE: Cloete) as well as the Technology and Human Resources for Industry program (THRIP – Grant number TP14081390585) of the South African National Research Foundation to SWPC.

## Additional information

### Funding

| Funder | Grant reference number | Author |
|---|---|---|
| Knut och Alice Wallenbergs Stiftelse | 2013.0129 & 2018.0138 | Charlie K Cornwallis |
| Vetenskapsrådet | 2017-03880 | Charlie K Cornwallis |
| Carl Tryggers Stiftelse för Vetenskaplig Forskning | 12: 92 & 19: 71 | Charlie K Cornwallis |
| Carlsbergfondet | | Mads F Schou |
| Jörgen Lindström stipendium | | Julian Melgar |
| Technology and Human Resources for Industry program | TP14081390585 | Schalk WP Cloete |
| Western Cape Agricultural Research Trust | 0070/000VOLSTRUISE | Schalk WP Cloete |

The funders had no role in study design, data collection and interpretation, or the decision to submit the work for publication.

### Author contributions

Julian Melgar, Conceptualization, Data curation, Software, Formal analysis, Validation, Investigation, Visualization, Methodology, Writing – review and editing; Mads F Schou, Data curation, Software,

Formal analysis, Supervision, Investigation, Methodology, Writing – review and editing; Maud Bonato, Zanell Brand, Anel Engelbrecht, Data curation, Investigation, Methodology, Writing – review and editing; Schalk WP Cloete, Conceptualization, Resources, Data curation, Funding acquisition, Investigation, Methodology, Project administration, Writing – review and editing; Charlie K Cornwallis, Conceptualization, Resources, Data curation, Software, Formal analysis, Supervision, Funding acquisition, Validation, Investigation, Visualization, Methodology, Writing – original draft, Project administration, Writing – review and editing

### Author ORCIDs
Julian Melgar ⓘ http://orcid.org/0000-0002-5718-8580
Schalk WP Cloete ⓘ http://orcid.org/0000-0002-4548-5633
Charlie K Cornwallis ⓘ http://orcid.org/0000-0003-1308-3995

### Ethics
This study was performed in strict accordance with the ethical permit AP/BR/O/SC14 granted by the Western Cape Government for Agriculture, South Africa. All of the animals were handled according to approved institutional animal care.

### Decision letter and Author response
Decision letter https://doi.org/10.7554/eLife.77170.sa1
Author response https://doi.org/10.7554/eLife.77170.sa2

## Additional files

### Supplementary files
• Supplementary file 1. Supplementary Tables A-S and R session information in html format. A: The composition of breeding groups observed in the Karoo National Park. B: Group size effects on the number of eggs produced by males. C: Group size effects on the number of eggs produced by females. D: Group size effects on the number of chicks produced by males. E: Group size effects on the number of chicks produced by females. F: Group size effects on the time nests were incubated. G: The effect of nest incubation on hatching success. H: Group size effects on the amount of time males spent incubating. I: Group size effects on the amount of time females spent incubating. J: Group size effects on the interruptions to incubation. K: The effect of the disparity in incubation on the number of interruptions. L: The effect of the disparity in incubation between males and females on % of eggs broken. M: The effect of the % of eggs broken on hatching success. N: Group size effects on the number of eggs produced by groups. O: Group size effects on the number of chicks produced by groups. P: The effect of average within-group relatedness on the number of chicks produced by males. Q: The effect of average within-group relatedness on the number of chicks produced by females. R: Sample size of experiment and summary statistics of reproductive success and incubation. S: Overview of replacements and removals of individuals in experimental groups.
• Transparent reporting form
• Source code 1. Contains the R code used for analyses.

### Data availability
All data and code are available at the open science framework: https://doi.org/10.17605/OSF.IO/4ZTFN.

The following dataset was generated:

| Author(s) | Year | Dataset title | Dataset URL | Database and Identifier |
|---|---|---|---|---|
| Cornwallis CK, Melgar J, Schou MF, Bonato M, Brand Z, Engelbrecht A, Cloete SWP | 2022 | Experimental evidence that group size generates divergent benefits of cooperative breeding for male and female ostriches | https://doi.org/10.17605/OSF.IO/4ZTFN | Open Science Framework, 10.17605/OSF.IO/4ZTFN |

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

## Appendix 1

### Supplementary analyses

We present two analyses in the supplementary materials (*Figure 2—figure supplement 3*; *Supplementary file 1* a4 and S15) that are not discussed in the main text, but may provide useful information to some readers. These analyses examine the effects of group composition on the total number of eggs (*Supplementary file 1n*; *Source code 1*: M13) and chicks (*Supplementary file 1o*; *Source code 1*: M14) produced by groups as opposed to the per individual measures of reproductive success presented in the main text. Finally, we verified that our analyses of group size effects on individual measures of reproductive success (Specific analyses section 'Testing how group size and the need for parental care influences male and female reproductive success') were not influenced by levels of relatedness within groups. Average relatedness between same sex individuals was calculated from a nine-generation pedigree generated from pair breeding adults (see *Schou et al., 2022*, for details). Average relatedness was logit transformed prior to analyses due to being bounded between 0 and 1. For males, model M3 was re-run including average relatedness between males and its interaction with care as fixed effects (*Supplementary file 1p*; *Source code 1*: M15). Data were restricted to groups with three males. For females, model M4 was re-run including average relatedness between females and its interaction with care as fixed effects (*Supplementary file 1q*; *Source code 1*: M16). Data were restricted to groups with more than one female. There was no effect of average relatedness between males or between females on the number of chicks individuals produced (*Supplementary file 1p & r*).

