## [Editor Report]

This article should be of interest to researchers working on animal behaviour and the evolution of cooperation. It experimentally investigates the effect of differences in group size and group composition on reproductive behavior, using an impressive sample of semi-wild populations of ostriches. Overall, the article offers a valuable analysis of the breeding ecology of ostriches and may inspire similar empirical work on other systems, examining cooperation, group living and breeding ecology.

---

## [Decision Letter]

**Decision letter after peer review:**

Thank you for submitting your article "Experimental evidence that animal societies vary in size due to sex differences in cooperation" for consideration by *eLife*.

Your article has been reviewed by two peer reviewers, and the evaluation has been overseen by a Reviewing Editor and Christian Rutz as the Senior Editor. The following individual involved in review of your submission has agreed to reveal their identity: Ralf H J M Kurvers (Reviewer #2).

The reviewers have discussed their reviews with one another, and the Reviewing Editor has drafted this decision letter to help you prepare a revised submission.

Essential revisions:

Apart from addressing the numbered essential revision requests below, please provide point-by-point responses to the reviewers' comments, and make changes to the manuscript accordingly.

1) The title is too broad and overstated considering the work done, and does not fully reflect what is presented in the paper. Reviewers suggest including an indication of the study species or at least group and a focus on the actual findings, limiting possible interpretations of the findings.

2) The introduction needs more structure, separation from methods, and clear delineation of hypotheses, aims, and expected results. Please provide the scientific background that led to this research and reference previous efforts.

3) Please separate results from the discussion as indicated by reviewers below. The discussion needs tempering of some of the claims and more discussion of alternative explanations due to some of the study limitations (e.g. no genetic data, see 7).

4) Overall. there is a need for more context with regard to ostrich biology, including typical group sizes, stability, and potential environmental effects. Please see reviewer comments below.

5) The issues of possible relatedness among the group members and individual reproductive success and how these may affect interpretations from group level measures used, should be discussed and alternative explanations discussed.

6) Due to potential confounds introduced by the experimental manipulations, please indicate whether optimal group size can actually be found in the dataset on natural variation in group size and group composition can be explored.

Finally, please note that *eLife* has recently adopted the STRANGE framework, to help improve reporting standards and reproducibility in animal behaviour research. In your revision, please consider scope for sampling biases and potential limitations to the generalisability of your findings:

https://reviewer.elifesciences.org/author-guide/journal-policies

https://doi.org/10.1038/d41586-020-01751-5

*Reviewer #1 (Recommendations for the authors):*

"Experimental evidence that animal societies vary in size due to sex differences in cooperation" – The title is too broad considering the work done and does not fully reflect what is presented in the paper. I suggest including an indication of the study species or at least group.

L21 "competition increases" -> not only competition, but also synchrony, which has a huge impact in group cohesion.

Introduction -> In general, the introduction is very short, especially considering that one paragraph is more methods than introduction. In my opinion, the aims are not clearly defined and not a working hypothesis or expected results are indicated.

L54-55 "fluctuating ecological conditions may shift the optimal size of groups over time and space"-> In some species, group size may even change with the habitat being used in a particular time. Is this the case for ostrich? How stable are the groups?

L69-73 "Males and females can also differ in their optimal group sizes due to divergent reproductive interests (Davies, 1989; Lessells, 2012; Trivers, 1972; Wong et al., 2012), but understanding how these effects combine to determine group size variation is difficult without experimental manipulations." -> I must say that I'm very sceptical about the need for experimental experiences to understand the effects of group size or determination of optimal group size. Most of all because the environmental factors have a huge impact on the optimal group size, that usually are not considered in experimental manipulations. I also would like to understand how stable the ostrich groups are usually, and why there is no contextualization to the effects of group size in the synchrony since that is one of the major costs that came with the increase of group size.

L87 "quantified group sizes in wild populations" -> Would be interesting if you compare your results for the experimental manipulation with an observational study in nature, using observations in the wild populations to compare and validate your experimental results. This would be a major and innovative result.

L91-91 "one to three" -> I'm not sure if I understood correctly, but in the figures, only 1 or 3 males appear represented. Can you explain please?

L100 – 103 "We collected data on the number of chicks produced over a period of five months where eggs were artificially incubated, and two months where eggs were naturally incubated (data on the number of eggs produced are presented in the supplementary material: Figure S1, Tables S2-S3)." -> The entire paragraph feels more like methods than like an introduction. The introduction is in fact too short and there is no background information on what is known about the breeding success of ostrich in nature, their group sizes, and whether or not other works on this subject were already done.

L-110 – 113 "This is similar to the …. across populations (Bertram, 1992; Kimwele and Graves, 2003; Magige et al., 2009)." -> This is not a result, is a discussion.

L110 "one to six females and one to three males" -> please indicate the mean number of males and females with their respective SE or CI. How many groups in the wild were observed to achieve this result?

L 164 – 170 "Limited opportunities for cooperation over incubation may constrain the reproductive… groups of intermediate size." -> Seems more discussion than results.

L189 – 195 "A lack of coordination… shaping the composition of cooperative breeding groups." -> is discussion, not results.

L283 "one or three males" – why not two males? there is any biological reason?

*Reviewer #2 (Recommendations for the authors):*

Title

I think the title is a little too far reaching given the results. First of all, why call it animal societies, rather than groups. More importantly, given the experimental results were done in captivity, a direct transfer of insights to wild conditions merits caution, given several other factors may determine group size in the wild. So I would change the title to better describe the actual findings, and not possible interpretations of the findings.

Abstract

L. 22: "variation in groups", should this be "variation in group size"?

I would add to the abstract that experimental manipulations were done in captivity to not oversell results.

If possible, I would add the used group sizes for males and females.

Intro

L. 40 I think one could add also here the benefits of collective defense, which is typically thought to be also an important reason for grouping.

L. 58 "Alternatively, assumptions about the way the cost and benefits change with group size may need revising". I do not fully understand this line of reasoning, given what has been said before. I fully agree that one should interpret correlations observed in the field with caution, and experimental manipulation is needed to separate cause and effect, but I do not see how this is necessarily at odds with the descriptions given in L. 53-56.

The point that drawing conclusions from observational data in the field due to a range of factors, is a very important one, and this can be even made a little stronger I think, by e.g., emphasizing the point of self-assortment a bit more (see e.g. intro in https://www.nature.com/articles/s42003-020-01597-7).

I would have expected a few more studies in the introduction reporting how group size determines fitness in the wild. From these one can then also make the point more forcefully that experimental manipulation is needed.

Also, the issue of sex differences in optimal group size may deserve a little more attention. It is a major focus of the research, but now only mentioned in passing in the introduction, and not really worked out (L. 69-71).

Another reason that optimal group sizes are not observed in the wild is that individuals are not free to choose which group to join, and hence may end up in suboptimal conditions.

In the introduction, it did not became clear to me whether it is known if ostriches breed in groups with related individuals. This would have repercussions how much we expect individuals to contribute to incubation. Also, for the experiments, it was not clear to me whether it could be verified whether individuals in groups were related or not.

Results

L. 108 Were the observed groups all breeding groups, or could this also be groups that were not breeding (and may never breed), e.g., siblings being together. It seems that "groups" and "breeding groups" are being equated in the manuscript, but that individuals are seen together is not always a guarantee they actually breed. Maybe it is in this system, but then please clarify.

L. 108 / Figure 1C though this is a nice figure, it is really hard to get an overall idea of the range of group size, and the sex ratio across group size. So it is really unclear if the typical range of males and females is indeed the one used for the experimental manipulations. I suggest to include a figure (at least in the supplement), showing on the x-axis group size and on the y-axis sex ratio, with size of the dots indicating number of observed groups at each point. And possibly above the figure, two histograms showing frequency distribution of nr of females and males across the range, for a quick visualization of the naturally occurring ranges of females and males.

L. 148 "that in multi-male and multi-female groups, individuals frequently shared incubation." It is possible that also when there were single male and/or single females in groups, there was sharing of incubation. Why constrain this sentence to multi-male / multi-female groups?

L. 149 "This led to the total time eggs were protected increasing". If I understand correctly, the terms nest protection and incubation are actually the same. If so, please only use one term (incubation favorable I think), also when describing the statistical models on this in the method section (L. 430-464). If not, please explain in the methods how "time spent protecting eggs" was defined as I could not find this in the methods.

L. 152 "hatching success". Please clarify what exactly this measure means (i.e., y-axis of Figure 3B). I could not find this in the method section. Was this the percentage of total laid eggs which hatched?

L. 155-156. And what about females? Please also report how the statistical relationship looked like for females.

L. 172 Please explain whether all these interruptions shown in Figure 4A were actually males interrupting females. Could it also be e.g. males interrupting males (aggression)? In the methods it is reported that sex and identity during interruptions were noted, but not clear to me what is reported here.

L. 189-190 "A lack of coordination between males and females over the timing of mating and incubation therefore appears to explain..." An alternative explanation is that in groups with multiple males, some males may not have copulated yet with one or more females, and/or they are uncertain whether the eggs are their own. This is not something that is really captured under "lack of coordination". This merits some more attention in the discussion, also the inverse U shaped pattern could be explored a bit more in the discussion.

Discussion

I think the discussion could use a bit more work, it is very short and missing some important points.

Some alternative explanations would deserve some more attention I think. For example, collective nest defense against predators can play a big role in determining the nesting success of ground breeding birds. I am not familiar with this aspect in ostriches, but a discussion on this issue seems relevant. In enclosures the impact of this selective force is weakened, so this may merit some caution when extrapolating the results to the field.

Individuals are not aiming to maximize their mean group level fitness, but their own fitness. A more thorough discussion on possible skewness in reproductive success in this system, and what the possible repercussions are for the presented results would be helpful.

L. 247-248 Some of the observed variation in breeding success may (or may not) come from between group differences in level of relatedness among group members or breeder quality, and even ecological conditions (i.e., quality of the enclosure), so I would be a bit cautious with drawing this conclusion. I think the experiments are very well done, and give a very meaningful contribution to the literature on cooperative breeding, but I would advise against overselling some of the claims.

Given the results, what would now be the prediction of how females versus males are expected to differ in optimal group size/group composition? And can this prediction be tested in the empirical data?

Methods

L. 255-257. Perhaps it can be added here whether (and why) November is a good time to perform observations to observe breeding groups (and not non-breeding groups).

L. 283 "one to six females". I would explicitly mention the used group sizes ("1, 2, 4 or 6 females").

L. 286. Table S16. The Supplement has been prepared with great care, which is very good. This table could be improved if actually the number of groups per year are shown. Currently, it is separated out by sex, but one cannot see which exact group compositions (that is combination of number of females and males per group) are tested when. In this Table it also becomes apparent that some treatments (e.g., groups with 6 females) were only done in 2015 and later, which is from an experimental design perspective far from ideal. Especially as from 2015 onward another procedure was used to determine hatching success under care conditions. This merits some more explanation and discussion. Why was this design so unbalanced and could this jeopardize the main results? Also, why was the sample size of groups with 4 females so much lower and suddenly stopped? It looks like midway the group sizes were adjusted.

L. 290 "replaced by a new individual". I would add "off the same sex" for clarification.

L. 305 "had to care for offspring". Should this not be "had to care for the brood"? It is not about the caring of offspring, but of the brood I think.

L. 324-326 + L. 352-353. If I understand it correctly, this procedure resulted in 2 values for each group, one for mean female success, and one for meal male success. Could this be added for clarification? One other way to interpret this, is that each individual in the group from the same sex received the same score, and these values were used for the modeling, which would create non-independence issues.

L. 332. "during this period". It was not entirely clear to me if incubation behavior was only scored when there were actual eggs present, or also when there were no eggs present (in which case I presume there is no incubation needed). Perhaps this can be clarified.

L. 383 "Parameter estimates (Ulster University) for fixed effects were calculated using posterior modes and are reported from models that included all terms of the same order and lower". Why not report the estimates from the full models (i.e., including all interactions)?

L. 426. Also, three-way interaction was included in the model according to the Tables, I would add this information in the description here (same for other sections).

L. 431 "The response variable was the number of observation minutes birds were sitting on nests versus the number of observation minutes nests were exposed." This description is a bit vague. What does "versus" mean? This almost sounds like there were two response variables (multivariate analyses?). Please clarify. Also L. 457.

L. 441 "Year" should probably be "year".

L. 460 Why was individual identity not nested in group identity? It was a nested design.

Figure 2. I would suggest adding the raw observed values as well, to give a better idea of the spread of the values.

Figure 3. Does "nest protection" and "incubation" actually mean the same thing? If yes, please use one term, if not, please explain how nest protection was measured.

Figure 3C I wonder if it is worth showing this separately for groups with 1 or 3 males. One would expect this line to be steeper when there is only one male present, than when 3 males are present, as in the 1 male groups, there is a larger relative change in group size moving from 1 to 6 females.

L. 534 "To understand natural variation in group size". I would be a bit cautious with using the results to explain natural variation in group size observed in the wild. For 3 main reasons. First, there are other reasons we may expect groups in the wild to vary which have not been controlled for. Second, within the experiments, other reasons may have played a role which drive between group variation. Third, individuals in the wild are not aiming to maximizing their mean hatching success per sex (as used as a measure here). So all of this should warrant caution with making strong claims about natural variation in group size and I would tone this down here, and in other places in the manuscript.

---

## [Author Response]

Essential revisions:Apart from addressing the numbered essential revision requests below, please provide point-by-point responses to the reviewers' comments, and make changes to the manuscript accordingly.

We are very grateful to the editors and reviewers for their positive and constructive feedback. Their input has been extremely valuable in revising the paper, which we now feel is substantially improved. Below are the details of the specific changes we have made to address their comments. Line numbers in word documents with track changes are sometimes not reproducible across computers. All line numbers therefore refer to the pdf version of the revised manuscript.

1) The title is too broad and overstated considering the work done, and does not fully reflect what is presented in the paper. Reviewers suggest including an indication of the study species or at least group and a focus on the actual findings, limiting possible interpretations of the findings.

We have now changed the title so that it reflects the results of the study more accurately and have included the study species.

2) The introduction needs more structure, separation from methods, and clear delineation of hypotheses, aims, and expected results. Please provide the scientific background that led to this research and reference previous efforts.

We have heavily revised the introduction to broaden the scientific background, clarify our aims, and incorporate additional literature. Given that the results are presented before the methods, we have retained a brief overview of our experimental approach at the end of the introduction as we believe this improves readability.

3) Please separate results from the discussion as indicated by reviewers below. The discussion needs tempering of some of the claims and more discussion of alternative explanations due to some of the study limitations (e.g. no genetic data, see 7).

We have revised the results and Discussion sections. We have moved some parts of the results to the discussion, which we have toned down and extended according to the specific reviewer’s comments. We have also included analysis of genetic parentage data from 3227 offspring, which verifies that measures of reproductive success we used in this study accurately capture the reproductive benefits for individuals (correlation between genetically determined parentage and measures used in analyses: *R* > 0.95. Figure 1—figure supplement 3 and methods section “Genetic parentage analysis”).

4) Overall, there is a need for more context with regard to ostrich biology, including typical group sizes, stability, and potential environmental effects. Please see reviewer comments below.

We have incorporated a more extensive description of relevant ostrich biology that includes the literature of what is known about the specific details requested by the reviewers.

5) The issues of possible relatedness among the group members and individual reproductive success and how these may affect interpretations from group level measures used, should be discussed and alternative explanations discussed.

This is an important point that we did not previously deal with. This is partly because the evidence from wild populations suggests that individuals do not preferentially associate with relatives (Kimwele et al., 2003 Molecular Ecology), although the evidence is limited. In our experiments, relatedness between adults in groups was known. We have now included extra analyses to show that average relatedness within groups did not influence the average reproductive success of individuals in groups (Supplementary Files 1p and 1q).

6) Due to potential confounds introduced by the experimental manipulations, please indicate whether optimal group size can actually be found in the dataset on natural variation in group size and group composition can be explored.

The apparent absence of an optimal group size (high variability in group size) in wild populations of ostriches, and other cooperative breeders, is one of the main premises for this manuscript. The results of our experimental study mirror what is observed in nature: there seems to be no overall optimal group size and this is expected because the benefits of cooperation and costs of competition associated with being in groups of different sizes differ for males and females.

We have clarified in the introduction that a key motivation for our study is explaining why the composition of cooperative breeding groups is highly variable in nature. We have also added a section to the discussion outlining the advantages of combining (1) experiments of captive populations, where entire social groups can be manipulated but ecological context may be lacking, with (2) observational studies of wild populations, where ecological conditions can be studied but manipulations of entire groups are difficult.

Finally, please note that eLife has recently adopted the STRANGE framework, to help improve reporting standards and reproducibility in animal behaviour research. In your revision, please consider scope for sampling biases and potential limitations to the generalisability of your findings:https://reviewer.elifesciences.org/author-guide/journal-policieshttps://doi.org/10.1038/d41586-020-01751-5

In accordance with *eLife* guidelines we have now added two brief statements: one in the Materials and methods section evaluating the ‘STRANGEness’ of the ostrich populations studied (lines 340-343), and one in the Discussion outlining how potential biases may limit the generalizability of our findings (lines 303-318).

Reviewer #1 (Recommendations for the authors):"Experimental evidence that animal societies vary in size due to sex differences in cooperation" – The title is too broad considering the work done and does not fully reflect what is presented in the paper. I suggest including an indication of the study species or at least group.

We have changed the title and included the study species.

L21 "competition increases" -> not only competition, but also synchrony, which has a huge impact in group cohesion.

We agree that synchrony and group cohesion between individuals can change with group size. We have modified the first sentence of the abstract accordingly, which reads “Cooperative breeding allows the costs of parental care to be shared, but as groups become larger such benefits often decline as competition increases and group cohesion breaks down.”

Introduction -> In general, the introduction is very short, especially considering that one paragraph is more methods than introduction. In my opinion, the aims are not clearly defined and not a working hypothesis or expected results are indicated.

The introduction has been extended. An extra paragraph has been added that provides more context to the study (lines 77-92), other paragraphs have been edited (lines 70-75; 97-104; 110-128), and the hypotheses we test have been more explicitly stated (lines 132-136).

L54-55 "fluctuating ecological conditions may shift the optimal size of groups over time and space"-> In some species, group size may even change with the habitat being used in a particular time. Is this the case for ostrich? How stable are the groups?

This is an important point. Understanding why groups vary in size and composition within the same environment during the same time is one of the main motivations for our paper. Observational studies of wild populations have confirmed that this is the case for ostriches (Sauer and Sauer, 1966; Bertram, 1992), just like many other cooperative breeders (Rubenstein and Abbott, 2017), making them a suitable study species. Unfortunately, individually marking and following ostriches in the wild is very difficult and has not yet been achieved. Consequently, little is known about the stability of wild breeding groups (but see Bertram 1992 for some discussion).

We have added extra details of what is known, and not known, about the breeding biology of ostriches (lines 116-128; 346-348).

L69-73 "Males and females can also differ in their optimal group sizes due to divergent reproductive interests (Davies, 1989; Lessells, 2012; Trivers, 1972; Wong et al., 2012), but understanding how these effects combine to determine group size variation is difficult without experimental manipulations." -> I must say that I'm very sceptical about the need for experimental experiences to understand the effects of group size or determination of optimal group size. Most of all because the environmental factors have a huge impact on the optimal group size, that usually are not considered in experimental manipulations. I also would like to understand how stable the ostrich groups are usually, and why there is no contextualization to the effects of group size in the synchrony since that is one of the major costs that came with the increase of group size.

We have expanded our discussion of why experimental studies are needed, as well as their limitations (lines 287-296; 303-308). As reviewer 2 highlights, observational studies alone do not provide causal insight into the factors influencing group size. However, as reviewer 1 indicates here experimental studies can lack ecological context. Therefore, we believe both are needed. Experimental studies are currently lacking on large vertebrate cooperative breeders, but can be used to estimate the costs and benefits of living in different group sizes, independently of ecological factors. The results of such experimental studies can be used as a benchmark against which other data can be compared, such as observations of group sizes from wild populations subject to ecological factors. The discrepancies between experimental and observational data can then be used to infer the relative importance of social versus ecological factors in shaping social groups.

We have edited the introduction and discussion to clarify the advantages and limitations of experimental studies, and the insights that can be gained by combining them with observational studies on wild populations (lines 94-114; 287-296; 303-320).

L87 "quantified group sizes in wild populations" -> Would be interesting if you compare your results for the experimental manipulation with an observational study in nature, using observations in the wild populations to compare and validate your experimental results. This would be a major and innovative result.

This is a very good point and one of the reasons why we tried to combine observational and experimental data. We realise that the links between the two should have been more explicit in the previous version. The observational data was largely used to illustrate that spatially and temporally overlapping groups are variable, rather than explicitly comparing this data with the experimental results.

We have added an extra figure (Figure 1—figure supplement 1) to make it easier to compare group variation in the wild to the experimental data on the costs and benefits of living in different groups for males and females. We have also provided extra discussion of the reasons why these two differ slightly, tying it in with the importance of ecological factors (see response to L69-73 above. Lines 310-320).

L91-91 "one to three" -> I'm not sure if I understood correctly, but in the figures, only 1 or 3 males appear represented. Can you explain please?

Addressed. It should have read “1 or 3”.

L100 – 103 "We collected data on the number of chicks produced over a period of five months where eggs were artificially incubated, and two months where eggs were naturally incubated (data on the number of eggs produced are presented in the supplementary material: Figure S1, Tables S2-S3)." -> The entire paragraph feels more like methods than like an introduction. The introduction is in fact too short and there is no background information on what is known about the breeding success of ostrich in nature, their group sizes, and whether or not other works on this subject were already done.

We are grateful for these suggestions and have added more background to what is known about the breeding system of ostriches (lines 117-128). Given the results are presented before the methods, we prefer to keep some experimental details at the end of the introduction to aid readability.

L-110 – 113 "This is similar to the …. across populations (Bertram, 1992; Kimwele and Graves, 2003; Magige et al., 2009)." -> This is not a result, is a discussion.

This statement references empirical data that is directly relevant to assessing the results that follow. These references are also mentioned in the discussion, but not including them here would make it unclear if group size variation is a more general phenomenon across populations. We therefore prefer to keep this statement.

L110 "one to six females and one to three males" -> please indicate the mean number of males and females with their respective SE or CI. How many groups in the wild were observed to achieve this result?

Figure 1—figure supplement 1 has been added to make this clearer.

L 164 – 170 "Limited opportunities for cooperation over incubation may constrain the reproductive… groups of intermediate size." -> Seems more discussion than results.L189 – 195 "A lack of coordination… shaping the composition of cooperative breeding groups." -> is discussion, not results.

We appreciate the suggestions for improving the style. The results are complicated in places, and we feel having some sentences to introduce the rationale behind tests is needed (e.g. L164 -170 original version). However, we have moved other parts not required for clarity to the discussion (e.g. L189 -195 original version).

L283 "one or three males" – why not two males? there is any biological reason?

Due to limitations in the number of birds accessible for our experiments, and other experiments being conducted on the same population, not all combinations of male and female group sizes were possible (see lines 284-286 of original version, 372-374 new version).

Reviewer #2 (Recommendations for the authors):TitleI think the title is a little too far reaching given the results. First of all, why call it animal societies, rather than groups. More importantly, given the experimental results were done in captivity, a direct transfer of insights to wild conditions merits caution, given several other factors may determine group size in the wild. So I would change the title to better describe the actual findings, and not possible interpretations of the findings.

We have now changed the title so that it reflects the results of the study more accurately and have included the study species.

AbstractL. 22: "variation in groups", should this be "variation in group size"?

Addressed.

I would add to the abstract that experimental manipulations were done in captivity to not oversell results.

We agree and have changed it accordingly.

If possible, I would add the used group sizes for males and females.

We have added exact group sizes to the abstract.

IntroL. 40 I think one could add also here the benefits of collective defense, which is typically thought to be also an important reason for grouping.

For cooperative breeders, including ostriches, one of the key benefits of group living is cooperation over offspring care. The collective raising of offspring is a key differentiator from many other group living species and therefore requires its own explanation. This is why we focus on cooperation over offspring care. This is not to say that other benefits of living in groups, such as collective defence, is not important. Indeed, it is for many social species, not just cooperative breeders. Given we do not examine collective defence we feel it may mislead readers to mention it here, so instead include it in the discussion.

L. 58 "Alternatively, assumptions about the way the cost and benefits change with group size may need revising". I do not fully understand this line of reasoning, given what has been said before. I fully agree that one should interpret correlations observed in the field with caution, and experimental manipulation is needed to separate cause and effect, but I do not see how this is necessarily at odds with the descriptions given in L. 53-56.

We agree that these two lines of reasoning, natural variation and unrealistic assumptions about the costs and benefits of group living, are not at odds with each other. The phrase “Alternatively” was misleading. We have revised the text accordingly (lines 77-92).

The point that drawing conclusions from observational data in the field due to a range of factors, is a very important one, and this can be even made a little stronger I think, by e.g., emphasizing the point of self-assortment a bit more (see e.g. intro in https://www.nature.com/articles/s42003-020-01597-7).

We are very grateful for this suggestion and have modified this section in accordance with the reviewer’s suggestions (lines 94-114).

I would have expected a few more studies in the introduction reporting how group size determines fitness in the wild. From these one can then also make the point more forcefully that experimental manipulation is needed.

We have added references of studies on wild populations of cooperative breeding cichlids, dunnocks, alpine accentors and greater anis and further stressed the need for experimental manipulations (lines 70-75, 106-114).

Also the issue of sex differences in optimal group size may deserve a little more attention. It is a major focus of the research, but now only mentioned in passing in the introduction, and not really worked out (L. 69-71).

We very much agree and have expanded our introduction of this issue (lines 267-275).

Another reason that optimal group sizes are not observed in the wild is that individuals are not free to choose which group to join, and hence may end up in suboptimal conditions.

This is a good point, but unfortunately it was not possible to address this in our study. We therefore raise the issue in the discussion, rather than the introduction, to avoid misleading readers that we examined individual decision making over group membership (lines 316-318).

In the introduction, it did not became clear to me whether it is known if ostriches breed in groups with related individuals. This would have repercussions how much we expect individuals to contribute to incubation. Also, for the experiments, it was not clear to me whether it could be verified whether individuals in groups were related or not.

We have added information and references to what is known about relatedness in groups of ostriches (lines 122-123). We have also added analyses of within-group relatedness that show it does not influence our results (Supplementary Files 1p and 1q).

ResultsL. 108 Were the observed groups all breeding groups, or could this also be groups that were not breeding (and may never breed), e.g., siblings being together. It seems that "groups" and "breeding groups" are being equated in the manuscript, but that individuals are seen together is not always a guarantee they actually breed. Maybe it is in this system, but then please clarify.

Ostriches breed continuously throughout the year. It is difficult to fully ascertain if groups are breeding without locating nests, which is extremely challenging. Consequently, groups were classified as those containing multiple sexually mature males and females that can be easily ascertained from feather and skin colouration. These were distinct from other groups commonly observed, where some individuals were sexually immature or where only one sex was present. We have clarified in the methods how we classify breeding groups (lines 363-364).

L. 108 / Figure 1C though this is a nice figure, it is really hard to get an overall idea of the range of group size, and the sex ratio across group size. So it is really unclear if the typical range of males and females is indeed the one used for the experimental manipulations. I suggest to include a figure (at least in the supplement), showing on the x-axis group size and on the y-axis sex ratio, with size of the dots indicating number of observed groups at each point. And possibly above the figure, two histograms showing frequency distribution of nr of females and males across the range, for a quick visualization of the naturally occurring ranges of females and males.

This is a great suggestion. We have included these figures as Figure 1—figure supplement 1.

L. 148 "that in multi-male and multi-female groups, individuals frequently shared incubation." It is possible that also when there were single male and/or single females in groups, there was sharing of incubation. Why constrain this sentence to multi-male / multi-female groups?

We have adjusted this sentence accordingly (line 203-204).

L. 149 "This led to the total time eggs were protected increasing". If I understand correctly, the terms nest protection and incubation are actually the same. If so, please only use one term (incubation favorable I think), also when describing the statistical models on this in the method section (L. 430-464). If not, please explain in the methods how "time spent protecting eggs" was defined as I could not find this in the methods.

We have replaced nest protection with incubation throughout.

L. 152 "hatching success". Please clarify what exactly this measure means (i.e., y-axis of Figure 3B). I could not find this in the method section. Was this the percentage of total laid eggs which hatched?

The referee is correct, it is the percentage of total eggs laid that hatched. This has been clarified in the methods (lines 444-446).

L. 155-156. And what about females? Please also report how the statistical relationship looked like for females.

We have added the statistics for females.

L. 172 Please explain whether all these interruptions shown in Figure 4A were actually males interrupting females. Could it also be e.g. males interrupting males (aggression)? In the methods it is reported that sex and identity during interruptions were noted, but not clear to me what is reported here.

Many thanks for pointing out this was unclear. It was only males interrupting female incubation. Males interrupting other males incubating and females interrupting other females incubating is rare. We have clarified this in the methods (line 436).

L. 189-190 "A lack of coordination between males and females over the timing of mating and incubation therefore appears to explain..." An alternative explanation is that in groups with multiple males, some males may not have copulated yet with one or more females, and/or they are uncertain whether the eggs are their own. This is not something that is really captured under "lack of coordination". This merits some more attention in the discussion, also the inverse U shaped pattern could be explored a bit more in the discussion.

Thank you for the suggestion. Uncertainty over paternity and the possibility that some males may not have copulated with one or more females is also likely to occur in groups with 4 or 6 females. However, interruptions were low in these groups. Consequently, we have not included an explicit discussion of these issues, but have re-phrased the Results section (lines 247-249) and added to the more general discussion of sexual conflict over reproduction (lines 263-285).

DiscussionI think the discussion could use a bit more work, it is very short and missing some important points.

The discussion has been extended according to the reviewer’s recommendations.

Some alternative explanations would deserve some more attention I think. For example, collective nest defense against predators can play a big role in determining the nesting success of ground breeding birds. I am not familiar with this aspect in ostriches, but a discussion on this issue seems relevant. In enclosures the impact of this selective force is weakened, so this may merit some caution when extrapolating the results to the field.

The importance of ecological factors, including nest predation, is now discussed more extensively (lines 298-308).

Individuals are not aiming to maximize their mean group level fitness, but their own fitness. A more thorough discussion on possible skewness in reproductive success in this system, and what the possible repercussions are for the presented results would be helpful.

Our aim was to estimate the average reproductive benefit from being in groups of different size for any given individual, not the mean fitness of the group. We have included analyses showing that genetic measures of individual reproductive success, which take into account reproductive skew, are highly correlated (*R*>0.95) with the measures of reproductive success used in our main analyses. We have also clarified our aims in the introduction and explained why using the average reproductive success per individual per sex is appropriate (lines 148-156; 417-422; 470-582).

L. 247-248 Some of the observed variation in breeding success may (or may not) come from between group differences in level of relatedness among group members or breeder quality, and even ecological conditions (i.e., quality of the enclosure), so I would be a bit cautious with drawing this conclusion. I think the experiments are very well done, and give a very meaningful contribution to the literature on cooperative breeding, but I would advise against overselling some of the claims.

Thank you for the suggestion. We have added additional analyses showing that variation in relatedness within groups does not alter the conclusions of our study (Supplementary Files 1p and 1q). We have also toned down the language to avoid over overstating the conclusions that can be drawn from our results.

Given the results, what would now be the prediction of how females versus males are expected to differ in optimal group size/group composition? And can this prediction be tested in the empirical data?

The focus of most research to date examining variation in cooperative breeding groups has been on the influence of ecological conditions and relatedness. The main conclusion of our study is that variation in group size and composition can arise independently of these factors.

Our results provide an estimate of the group sizes where the reproductive success of the average male and female is maximised: For males, reproductive success is maximised in groups consisting of a single male and 4 or more females, and for females, reproductive success is maximised in groups that are larger than pairs and do not contain intermediate numbers of males and females (3 males and 3 females). These estimates can then be used as a “null expectation” to test predictions about the effect of other variables on group size, such as ecological factors. In this way, it might be possible to understand the relative contributions of social interactions and ecological conditions to variation in cooperative breeding groups.

We have revised the manuscript to draw stronger links between our experimental data and observations on wild populations and discuss how our results can be used in future studies (lines 310-320; Figure 1—figure supplement 1).

MethodsL. 255-257. Perhaps it can be added here whether (and why) November is a good time to perform observations to observe breeding groups (and not non-breeding groups).

We have added that November is during the middle of the breeding season for ostriches in South Africa (lines 334-337).

L. 283 "one to six females". I would explicitly mention the used group sizes ("1, 2, 4 or 6 females").

Addressed throughout the manuscript.

L. 286. Table S16. The Supplement has been prepared with great care, which is very good. This table could be improved if actually the number of groups per year are shown. Currently, it is separated out by sex, but one cannot see which exact group compositions (that is combination of number of females and males per group) are tested when. In this Table it also becomes apparent that some treatments (e.g., groups with 6 females) were only done in 2015 and later, which is from an experimental design perspective far from ideal. Especially as from 2015 onward another procedure was used to determine hatching success under care conditions. This merits some more explanation and discussion. Why was this design so unbalanced and could this jeopardize the main results? Also, why was the sample size of groups with 4 females so much lower and suddenly stopped? It looks like midway the group sizes were adjusted.

We have added group-composition to the table as suggested. Our experiments were extremely challenging to setup and had to accommodate multiple research projects on the same study population. The changes in study design over the years were because of balancing different study aims with the availability of birds and facilities. Although we acknowledge that this was not ideal, we believe that collating a larger, longer-term dataset was more valuable than restricting data to specific years that were more comparable. Importantly, numerous group sizes were represented throughout the entire study period allowing the impact of changes in design to be estimated. Overall, we found very little evidence that changes in design and hatching procedure had any effect (see random effect estimates of year in Supplementary Files 1b – 1m).

L. 290 "replaced by a new individual". I would add "off the same sex" for clarification.

Addressed.

L. 305 "had to care for offspring". Should this not be "had to care for the brood"? It is not about the caring of offspring, but of the brood I think.

Addressed.

L. 324-326 + L. 352-353. If I understand it correctly, this procedure resulted in 2 values for each group, one for mean female success, and one for meal male success. Could this be added for clarification? One other way to interpret this, is that each individual in the group from the same sex received the same score, and these values were used for the modeling, which would create non-independence issues.

This is correct, there were two values for each group, one for males and one for females. This is because we were interested in the reproductive returns for any given individual, rather than variation between individuals within groups. Non-independence is not an issue as there is only one value per sex per group, rather than multiple individuals of the same sex within the same group.

L. 332. "during this period". It was not entirely clear to me if incubation behavior was only scored when there were actual eggs present, or also when there were no eggs present (in which case I presume there is no incubation needed). Perhaps this can be clarified.

We monitored incubation behaviour only during the period when eggs were left in groups. If females laid eggs in nests after this point, then birds typically initiated incubation once a certain number of eggs had accumulated. If females did not lay eggs, then incubation did not occur. We have clarified this in the text (lines 433-435).

L. 383 "Parameter estimates (Ulster University) for fixed effects were calculated using posterior modes and are reported from models that included all terms of the same order and lower”. Why not report the estimates from the full models (i.e., including all interactions)?

Estimates are not valid if they are involved in higher order interactions. For example, main effects should not be interpreted in models where they are involved in interactions. We therefore report estimates where all terms of the same order are included (e.g. main effects are presented from models including all other main effects ect..). We are grateful to the referee for pointing out this was unclear and have clarified the text accordingly (lines 597).

L. 426. Also, three-way interaction was included in the model according to the Tables, I would add this information in the description here (same for other sections).

Addressed (lines 638-641, 652-654, 678-680, 694-696).

L. 431 "The response variable was the number of observation minutes birds were sitting on nests versus the number of observation minutes nests were exposed." This description is a bit vague. What does "versus" mean? This almost sounds like there were two response variables (multivariate analyses?). Please clarify. Also L. 457.

The % of time nests were incubated was modelled as a single binomial response variable. To account for variation in observation time, the number of minutes nests were incubated relative to the number of minutes nests were not incubated were analysed, as is standard practice for generalised linear models with binomial error distributions. We have removed “versus” and clarified our approach (line 674).

L. 441 "Year" should probably be "year".

Addressed.

L. 460 Why was individual identity not nested in group identity? It was a nested design.

Individuals were nested within groups. As each individual was only present in a single group each year and each group was only present in a single year, nesting is implicit in our mixed model specification of MCMCglmm.

Figure 2. I would suggest adding the raw observed values as well, to give a better idea of the spread of the values.

Thank you for the suggestion. We have revised Figures 2, 3 and 4, Figure 1—figure supplement 2, and Figure 2—figure supplement 4 to show raw data and adjusted the colour scheme to be clearer in black and white.

Figure 3. Does "nest protection" and "incubation" actually mean the same thing? If yes, please use one term, if not, please explain how nest protection was measured.

We have replaced nest protection with incubation throughout.

Figure 3C I wonder if it is worth showing this separately for groups with 1 or 3 males. One would expect this line to be steeper when there is only one male present, than when 3 males are present, as in the 1 male groups, there is a larger relative change in group size moving from 1 to 6 females.

Separate lines have now been included for groups with 1 and 3 males.

L. 534 "To understand natural variation in group size". I would be a bit cautious with using the results to explain natural variation in group size observed in the wild. For 3 main reasons. First, there are other reasons we may expect groups in the wild to vary which have not been controlled for. Second, within the experiments, other reasons may have played a role which drive between group variation. Third, individuals in the wild are not aiming to maximizing their mean hatching success per sex (as used as a measure here). So all of this should warrant caution with making strong claims about natural variation in group size and I would tone this down here, and in other places in the manuscript.

We have re-phrased this sentence and toned-down claims about explaining natural variation in group size.